# NOISE-AWARE GENERALIZATION: ROBUSTNESS TO IN-DOMAIN NOISE AND OUT-OF-DOMAIN GENERALIZATION

**Siqi Wang, Aoming Liu, Bryan A. Plummer**
Department of Computer Science
Boston University
{siqiwang,amliu,bplum}@bu.edu

## ABSTRACT

Methods addressing Learning with Noisy Labels (LNL) and multi-source Domain Generalization (DG) use training techniques to improve downstream task performance in the presence of label noise or domain shifts, respectively. Prior work often explores these tasks in isolation, and the limited work that does investigate their intersection, which we refer to as Noise-Aware Generalization (NAG), only benchmarks existing methods without also proposing an approach to reduce its effect. We find that this is likely due, in part, to the new challenges that arise when exploring NAG, which does not appear in LNL or DG alone. For example, we show that the effectiveness of DG methods is compromised in the presence of label noise, making them largely ineffective. Similarly, LNL methods often overfit to easy-to-learn domains as they confuse domain shifts for label noise. Instead, we propose Domain Labels for Noise Detection (DL4ND), the first direct method developed for NAG which uses our observation that noisy samples that may appear indistinguishable within a single domain often show greater variation when compared across domains. We find DL4ND outperforms DG and LNL methods, including their combinations, even when simplifying the NAG challenge by using domain labels to isolate domain shifts from noise. Performance gains up to 12.5% over seven diverse datasets with three noise types demonstrates DL4ND's ability to generalize to a wide variety of settings[1].

## 1 INTRODUCTION

Domain Generalization (DG) methods train models to generalize to unseen target domains by learning from multiple source domains (*e.g.*, Cha et al. (2022; 2021); Wang et al. (2023); Arjovsky et al. (2019); Kamath et al. (2021); Chen et al. (2023; 2024a); Rame et al. (2022); Lin et al. (2022)). However, these methods tend to ignore label noise, which appears naturally in many datasets (*e.g.*, Chen et al. (2024b); Xiao et al. (2015); Li et al. (2017c); Song et al. (2019)), including those used in DG benchmarks (Teterwak et al., 2025a). Despite this, limited prior work has explored the effect of label noise in DG settings (Qiao & Low, 2024; Seo et al., 2020), which simply evaluates existing methods without proposing solutions to improve robustness. A naive approach to mitigate label noise issues would be simply to combine DG methods with those from the Learning with Noisy Labels (LNL) literature (*e.g.*, Natarajan et al. (2013); Arpit et al. (2017); Song et al. (2022); Xia et al. (2022; 2023); Wei et al. (2022); Liu et al. (2021); Song et al. (2024); Cordeiro et al. (2023); Shen & Sanghavi (2019); Wang & Plummer (2024)). Generally speaking, LNL methods work by detecting potential noise and then either removing (Mirzasoleiman et al., 2020; Kim et al., 2021), relabeling (Karim et al., 2022; Li et al., 2023), or downweighting (Liu et al., 2020; 2023) these samples. However, as shown in Figure 1, separating distribution shifts due to noise from distribution shifts due to domain is challenging since they look similar according to feature similarity or analyzing loss values (an observation echoed in related work on out-of-distribution detection (Humblot-Renaux et al., 2024)).

---

[1] Code: https://github.com/SunnySiqi/Noise-Aware-Generalization

**How to Separate Label Noise from Distribution Shifts?**

Prediction: 0.99 Dog
Label: Dog
Loss: 0.01

**NAG Challenge:** Unable to distinguish solely based on predictions, losses or feature similarities.

Prediction: 0.51 Dog
Label: Dog
Loss: 0.67

**Distribution Shift** should be **Included** from training for generalization.

Similarity Score: 0.35

Prediction: 0.52 Dog
Label: Dog
Loss: 0.65

**Noisy Label** should be **excluded** from training to avoid overfitting.

Similarity Score: 0.37

Figure 1: A key challenge in NAG is being able to separate distribution shifts due to domain from shifts due to noise. However, in practice these samples can be hard to distinguish between each other, which makes mitigating the effect of label noise challenging as many LNL methods require some mechanism of detecting the noise (*e.g.*, Li et al. (2023); Karim et al. (2022); Zhao et al. (2024)). This represents a new challenge that only emerges when considering learning with noisy labels and domain generalization together, highlighting the need to explore NAG.

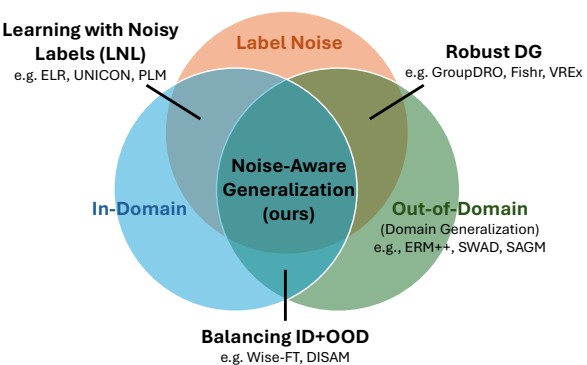

Figure 2: DG methods often ignore in-domain performance (*e.g.* Sagawa et al. (2020); Rame et al. (2022); Krueger et al. (2021); Zhang et al. (2024); Wortsman et al. (2022)), label noise (*e.g.* Teterwak et al. (2025b); Cha et al. (2022; 2021); Wang et al. (2023)), or both. LNL methods handle in-domain label noise, but not domain shifts (Liu et al., 2020; Li et al., 2023; Karim et al., 2022; Zhao et al., 2024). In contrast, NAG imitates real-world applications by requiring methods that do well on all data when trained on noisy data.

We refer to the intersection of DG and LNL as Noise-Aware Generalization (NAG), where the goal is to maximize both in-domain (ID) and out-of-domain (OOD) performance when training on noisy, multi-domain datasets. This makes NAG methods more practical as they can be applied to a wider range of settings than DG or LNL alone. In particular, there are three major components in Figure 2 we use to compare NAG to related tasks: 1) methods that are evaluated on ID performance, 2) methods that are evaluated on OOD performance, 3) methods that are evaluated when trained with label noise. The benchmarks used by method in prior work would cover two of these settings at most. For example, DG methods either ignore in-domain performance (*e.g.*, Sagawa et al. (2020); Rame et al. (2022); Krueger et al. (2021)) or label noise (*e.g.*, Teterwak et al. (2025b); Cha et al. (2022; 2021); Wang et al. (2023)). LNL methods handle noise, but do not consider domain shifts (Liu et al., 2020; Li et al., 2023; Karim et al., 2022; Zhao et al., 2024). However, it is common to deploy a model that should perform well on both ID and OOD data, and label noise is expected as it is often prohibitively expensive to collect perfectly clean data (*e.g.*, where expertise is required or annotators simply disagree (Chen et al., 2024b; Tanno et al., 2019; Vorontsov & Kadoury, 2021)).

We begin by exploring NAG by analyzing the challenges through experiments on a synthetic noise dataset, providing foundational insight for further research. In particular, we highlight the issues with developing LNL-style methods that aim to detect noise, partly highlighted in Figure 1, where identifying that distribution shifts in the training set are due to noise rather than domain shifts is

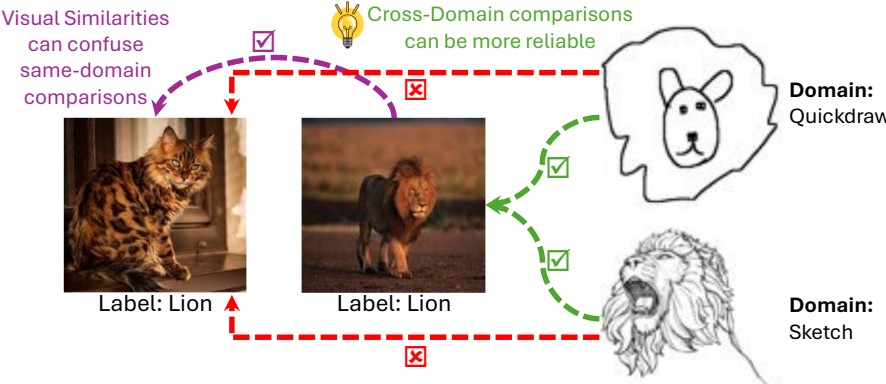

Figure 3: **Images in example embedding space.** Comparing images within the same domain (shown in purple) can rely on spurious features, such as similar colors, resulting in labels being incorrectly identified as clean. However, cross-domain comparisons (shown in red and green) must rely on intrinsic features, thereby providing a more accurate estimate of an image's label.

challenging. A naive solution to this problem would be to separate samples into their respective domains. In effect, this would theoretically reduce the problem to the traditional LNL task, where a method would identify noisy samples within a single domain. However, spurious features in some domains make them more challenging to detect.

For example, the two images labeled as "lion" in Figure 3 have similar colors, which a model may inadvertently use as most lions also have the same colors. Thus, separating them within the photo domain would be harder than across domains where they must rely on intrinsic, domain-agnostic features. This observation motivates our proposed method, Domain Labels for Noise Detection (DL4ND), which identifies noisy samples by extracting (class, domain) proxies from low-loss samples. Then we use these proxies for reliable noise detection through cross-domain comparisons. We show that DL4ND can improve performance over other LNL methods on its own, or can be combined with DG methods for further gains. Experiments with 12 state-of-the-art DG and LNL methods, along with 20 combination methods on three real and four synthesized noise datasets, show that DL4ND performs up to 12.5% better than prior work.

Our contributions are summarized below.
- We highlight the challenges of real-world datasets that exhibit both label noise and domain shifts in diverse fields, including web data (Fang et al., 2013) and biological imaging (Chen et al., 2024b).
- We investigate the underexplored setting we call Noise-Aware Generalization (NAG), which focuses on training a robust network under ID noise while ensuring good generalization to OOD data. We analyze the limitations of existing approaches and their naive combinations to provide insight into NAG and its challenges.
- We propose DL4ND, a novel noise detection method that can be used on its own or in combination with prior work in DG, showing a promising solution to NAG.

## 2 RELATED WORK

As discussed in the Introduction and summarized in Figure 2, NAG combines elements from both the Domain Generalization (DG) and Learning with Noisy Labels (LNL), producing new challenges not present when exploring each task alone. DG methods (*e.g.*, Gulrajani & Lopez-Paz (2021b); Li et al. (2017a;b; 2019; 2018a;b); Muandet et al. (2013); Teterwak et al. (2025b)) often focus on learning domain-invariant features (*i.e.*, aligning their representations). In contrast, LNL methods typically aim to minimize the effect of label noise by reweighting samples based on an estimate of cleanliness (*e.g.*, Scott (2015); Liu & Tao (2015); Menon et al. (2015); Patrini et al. (2017); Li et al. (2021); Zhang et al. (2021); Kye et al. (2022); Cheng et al. (2022); Liu et al. (2023); Li et al. (2022b); Vapnik et al. (2015); Yong et al. (2022)) or detecting and removing (or relabeling) the noisy samples (*e.g.*, Hu et al. (2021); Torkzadehmahani et al. (2023); Nguyen et al. (2020); Tanaka et al. (2018); Hou et al. (2025); Li et al. (2022a); Feng et al. (2022); Li et al. (2023); Song et al. (2024);

Xia et al. (2023); Cordeiro et al. (2023); Wei et al. (2022); Xia et al. (2022)). However, DG methods often ignore the effect of label noise and only evaluate on out-of-domain (OOD) generalization. In contrast, LNL methods study the effect of label noise on in-domain (ID) data, but ignore domain shifts. Thus, these methods may not generalize in real-world settings like explored by NAG, where (at least some) label noise is expected and models have to perform well on ID and OOD settings.

While there is limited DG work that evaluates based on ID and OOD performance (*e.g.*, Zhang et al. (2024); Wortsman et al. (2022)), these methods do not explore the effect of label noise. Similarly, prior work that evaluates existing DG method's performance with label noise do not develop new techniques for this setting (Qiao & Low, 2024; Seo et al., 2020), only exploring preexisting approaches. Some work that explores the intersection of Domain Adaptation (DA) and LNL proposes custom methods (*e.g.*, Shu et al. (2019); Han et al. (2023); Zuo et al. (2022); Zhuo et al. (2023); Feng et al. (2023); Yin et al. (2025)), but as DA is typically performed a single source domain, they do not have to learn to separate domain shifts from label noise (*i.e.*, they do not have to address the challenge in Figure 1). As such, many of the DA+LNL methods closely match those in the LNL literature (see Appendix B for examples), with similar limitations. Additionally, prior work in DG+LNL and DA+LNL are typically not evaluated on their in-domain performance. As our experiments show, these limitations mean that they do not generalize well to NAG, often performing on par with simple ERM (Gulrajani & Lopez-Paz, 2021a). Thus, exploring NAG is a necessary step to creating methods that generalize to real-world settings.

## 3 NOISE-AWARE GENERALIZATION (NAG)

Consider a multi-domain dataset $\mathcal{D}$ with $m$ source domains: $\mathcal{D} = \{\mathcal{D}_1, \mathcal{D}_2, \ldots, \mathcal{D}_m\}$, where each $\mathcal{D}_i = \{(x_{i,j}, \tilde{y}_{i,j})\}_{j=1}^{n_i}$ represents samples from domain $i$ with $x_{i,j}$ as the input and $\tilde{y}_{i,j}$ as the label, potentially noisy and the true label $y_{i,j}$ is unknown. The goal in NAG is to learn a featurizer $f_\theta(\cdot)$ parameterized by $\theta$ that performs well in all source domains $\{\mathcal{D}_i\}_{i=1}^m$ and generalizes to unseen domain(s) $\mathcal{D}_{target}$, despite the presence of label noise. For convenience in describing the equation in the rest of the section, we denote the domain of an input $x$ as $D(x)$ and its class label as $Y(x)$. We use $d(\cdot)$ to represent the cosine distance between feature embeddings.

To help better understand NAG, we perform a preliminary analysis using RotatedMNIST (Ghifary et al., 2015), chosen for its simplicity and clear feature structure. Note that we also validate these observations on more complex datasets in our experiments. We select four rotation angles (0°, 15°, 30°, and 45°) to represent different domains. Pairwise noise is introduced by randomly flipping 30% of the labels between four confusing digit pairs: (0, 6), (1, 7), (3, 5), and (4, 9). Our experiments use a ResNet50 (He et al., 2016) trained via ERM (Gulrajani & Lopez-Paz, 2021a).

### 3.1 CAN WE SEPARATE CLASS SHIFTS FROM DOMAIN SHIFTS?

Detecting samples that may stem from label noise is a key capability for LNL-style methods. However, as NAG also requires generalizing across domains, an important capability that distinguishes NAG from related work is being able to separate samples with shifts due to domain from shifts due to label noise during training (partially highlighted in Figure 1). However, measuring distributional differences between samples (*e.g.*, by computing feature distances between all sample pairs) can be computationally expensive. Thus, LNL methods often rely on having some canonical representation of a class to measure similarity of a particular sample (*e.g.*, Mirzasoleiman et al. (2020); Song et al. (2024); Zhao et al. (2024)), for example, by averaging the features of samples of that class. If shifts in domain were closer to this canonical representation than shifts in category, then we could successfully identify potential noise.

More formally, continuing the earlier notation, let us group samples in domain-class sets, *i.e.*,

$$G_{c,i} = \{x_j \mid Y(x_j) = c, \mathcal{D}(x_j) = i\}. \tag{1}$$

Let $\bar{g}_{c,i} = \frac{1}{|G_{c,i}|} \sum f_\theta(G_{c,i})$ denote the average of the set of learned features for group $G_{c,i}$. We assume the existence of a featurizer $f_\theta(\cdot)$ such that:

$$d(f_\theta(G_{c,\hat{i}}), \bar{g}_{c,i}) < d(f_\theta(G_{\hat{c},i}), \bar{g}_{c,i}), \text{ where } i \neq \hat{i} \text{ and } c \neq \hat{c}. \tag{2}$$

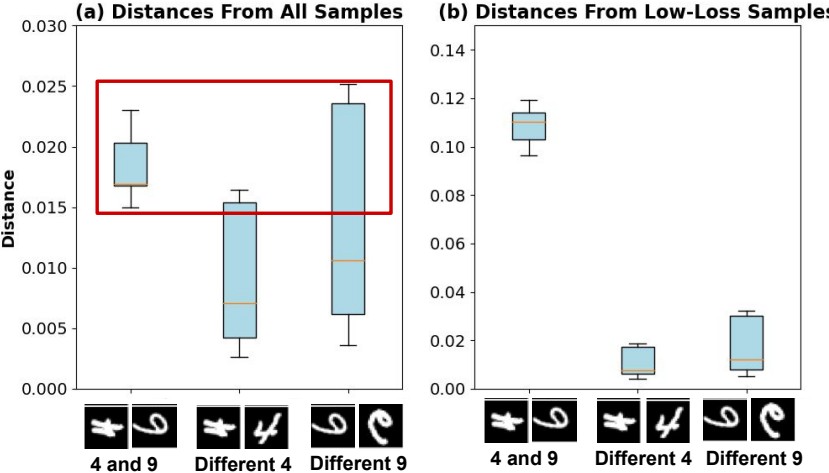

Figure 4: **Box plot of distance distributions between samples and their canonical class representation** $\bar{g}_{c,i}$. (a) Reports distances between training samples and an $\bar{f}_D$ computed by averaging all training samples in its class, domain group $G_{c,i}$. The red box highlights overlapping distributions, indicating the challenge of distinguishing samples with class and domain shift. (b) $\bar{g}_{c,i}$ is calculated from low-loss samples which shows shifts in class can be separated from shifts in domain. See Section 3.1 for additional discussion.

This suggests that we can perfectly separate class shifts from domain shifts if we can train a featurizer such that, for each sample, the distance to other samples within the same class (across different domains) is smaller than the distance to samples within the same domain (but different classes).

Figure 4 explores this possibility by showing the distribution of distances in the presence of both shifts in domain and shifts in class on RotatedMNIST (Ghifary et al., 2015). As seen in the red box in Figure 4(a), where $\bar{g}_{c,i}$ is computed over all samples in the group $G_{c,i}$, there is a significant portion of the shifts in category and domain that cannot be distinguished from each other (verifying the example in Figure 1 occurs in practice). Thus, Equation (2) is not satisfied and we cannot use this approach to identify noisy samples. However, as shown in Figure 4(b), we find that we can satisfy Equation (2) by creating $\bar{g}_{c,i}$ with samples we have high confidence are not noise. For example, by using only low-loss samples early in training (*i.e.*, before the model begins overfitting)that prior work has shown are typically clean (Liu et al., 2020; Choi et al., 2025). See further validation of these observations on more complex datasets in Section 5.

### 3.2 Is it really that important to satisfy Equation (2)?

As discussed in the previous section, if we are not careful about how we measure feature similarity between samples then we can end up in a situation like for samples in red box in Figure 4(a), where we simply can't distinguish between shifts due to class and shifts due to domain. However, one could argue that if we are unsure about these samples, we should simply mitigate their contribution to model training (*e.g.*, by downweighting their loss or removing them entirely). *I.e.*, if we can still learn the decision boundary between classes without them, then these samples are unnecessary.

To quantify sample importance to the final decision boundary, we trained an SVM on RotatedMNIST and treat the resulting support vectors as the "important" samples. We find that over 20% of the support vectors fall within the red box region of Figure 4(a). Thus, as we show in Section 5, removing their contribution during training has a significant effect on the final decision boundary.

### 4 Domain Labels for Noise Detection (DL4ND)

In Section 3 we gave a formal definition of NAG and also provided an analysis that lead to a means of separating domain shifts from category shifts by using a small set of samples to create a proxy to represent the canonical features of a particular category. If all datasets were as simple as Rotat-

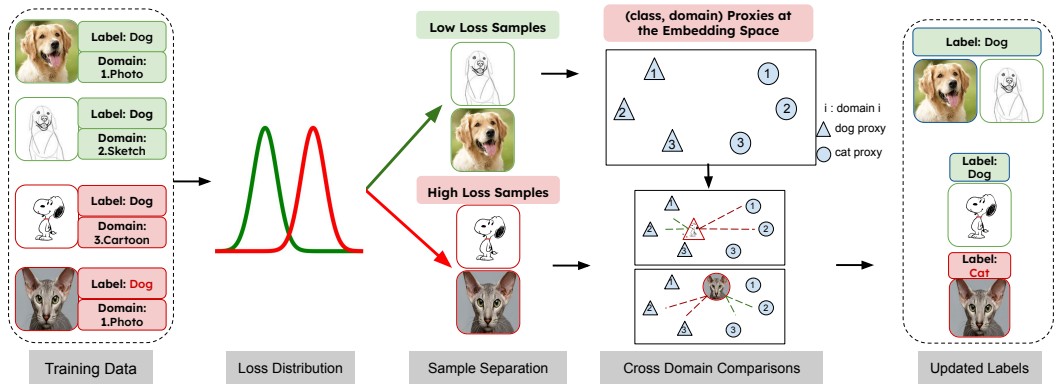

Figure 5: **DL4ND Framework**. After a brief warmup stage, the first step in DL4ND is to split samples into low-loss and high-loss groups using a Gaussian Mixture Model (GMM) based on the loss distribution. The low-loss sample's labels are frozen and used to generate (class, domain) proxies $\bar{g}_{c,i}$. High-loss samples are relabeled using the proxies with Equation (3). Then training resumes with the updated labels. Note that in all stages, a DG method (*e.g.*, ERM++ (Teterwak et al., 2025b), SAGM (Wang et al., 2023), SWAD (Cha et al., 2021)) can also be used in combination with DL4ND. See Section 4.2 for additional discussion.

edMNIST, this might be sufficient to solve the NAG task. However, in more complex datasets there can be many confounding factors, such as due to biases in a source domain (see Figure 3), that can confuse a model. Section 4.1 aims to reduce the impact of these spurious features via DL4ND, our proposed approach for detecting noise by using cross-domain comparisons. Section 4.2 describes how we fit our DL4ND noise detection method into our full NAG framework. See Figure 5 for an illustration summarizing our approach.

## 4.1 DETECTING NOISE WITH CROSS-DOMAIN COMPARISONS

As illustrated in Figure 3, noisy samples may exhibit strong visual similarity to their incorrect noisy labels within a given domain. This "visual similarity" often arises from spurious features, such as background or color, which are domain-dependent and may not persist across different domains. As we will discuss further below, these issues persist even if we were to simplify the NAG problem by using domain labels to keep training samples in their own domain, *i.e.*, to perform single-domain comparisons to identify noise. These observations lead us to formulate a hypothesis: can cross-domain comparisons provide a stronger signal for a sample's true class than single-domain methods?

Let us perform a thought experiment with the example in Figure 3. Since the lion images in the photo domain have a quintessential set of colors used for a typical lion, its reasonable that a model might use these correlations in the color space to identify if an image is a lion. After all, if an image contained largely black pixels, then the model knows it is unlikely to be a lion in this domain. However, other domains may not have the same color bias, *e.g.*, the sketch and draw images in Figure 3. Thus, comparing images from these domains to the photos requires the model to rely on more lion's intrinsic features that generalize across domains. Motivated by this observation (which we empirically validate in Section 5), our DL4ND creates a new label $\hat{y}_i$ for sample $x_i$ identified as potential noise by finding the closest class representation from another domain $\bar{g}_{c,\hat{i}}$, *i.e.*,:

$$\hat{y}_i = \underset{\forall g_{c,\hat{i}}}{\arg\min}\, \mathrm{d}(f_\theta(x_i), \bar{g}_{c,\hat{i}}), \text{ where } i \neq \hat{i}. \tag{3}$$

Note that this does not mean that $\hat{y}_i \neq y_i$ in all cases, as some $x_i$ may have their labels judged as correct according to Equation (3). Next we discuss how Equation (3) is used in our full framework.

## 4.2 DL4ND FRAMEWORK

Prior work showed noisy labels have minimal effect on early stages of training (Liu et al., 2020). In this stage the concept of a category is still being formed, often using general, easy-to-learn features.

Thus, LNL methods like DL4ND are typically applied in later stages where a model may start overfitting to noisy labels. This means we begin training using ERM (Gulrajani & Lopez-Paz, 2021a), or any other DG method, which is used in all stages of training. After this warmup step, we first create our class proxies using the low-loss samples inspired by the discussion from Section 3.1.

Specifically, rather than introducing a hyperparameter that manually controls the loss threshold, we assume the loss distribution is separated by a Gaussian Mixture Model (GMM) with two clusters. Samples belonging to the low-loss cluster serve as proxies, while high-loss samples require label updates through cross-domain comparisons. Low-loss samples are grouped by both domain and class (as shown in Equation (1)). This means each (domain, class) pair has its own proxy representation, computed as the average feature of all low-loss samples in the same (domain, class) group. We assume these low-loss samples have clean labels and are kept frozen during training.

High-loss samples are relabeled using the cross-domain comparisons defined in Equation (3). Thus, DL4ND performs label refinement during training, requiring no additional data or learning overhead. This relabeling can be done periodically, as done in prior work Karim et al. (2022); Li et al. (2023). However, our experiments show that labeling only once can improve label quality. For example, Table 1 shows DL4ND on RotatedMNIST increased label accuracy from 75% to 98%, resulting in a 10% boost in both ID and OOD accuracy.

Table 1: Top-1 accuracy on RotatedMNIST (Ghifary et al., 2015) with 30% asymmetric label noise. The baseline method creates $\bar{f}_D$ using all samples from its group $G_{c,i}$ with single-domain rather than cross-domain comparisons. See Section 4.2 for discussion.

| Method | Label Acc. | ID | OOD |
|---|---|---|---|
| Baseline | 75.7 | 87.7 | 87.9 |
| DL4ND (ours) | 98.1 | 98.1 | 97.8 |

## 5 EXPERIMENTS

**Datasets.** We use three real-world datasets (VLCS (Fang et al., 2013), CHAMMI-CP (Chen et al., 2024b), and PACS (Li et al., 2017a)) and three additional synthetic noise datasets (Office-Home (Venkateswara et al., 2017), TerraIncognita (Beery et al., 2018), and DomainNet (Peng et al., 2019)) to supplement our RotatedMNIST experiments reported earlier. Appendix C discusses our noise types, which supplements the results using asymmetric noise on RotatedMNIST with real-world highly noisy dominant noise (Wang & Plummer, 2024) (*e.g.*, CHAMMI-CP) with both real and synthetic asymmetric and symmetric noise (VLCS and PACs for real noise, and DomainNet, Officehome, and TerraIncognita for synthetic). These datasets provide results over a ride range of applications, *e.g.*, VLCS and DomainNet are typical web images, CHAMMI-CP contains cell images, and TerraIncognita contains images from wildlife cameras.

**Metrics.** All of our datasets report top-1 classification accuracy on both ID and OOD data. In some experiments we also report the average of these two metrics to measure overall NAG performance.

**Experimental setup.** As shown in Figure 2, there are three components to evaluating NAG: in-domain (ID) performance, out-of-domain (OOD) generalization, and robustness to noise. Our experiments are designed to evaluate each component by reporting both ID and OOD accuracy using a "leave-one-out" protocol used by domain generalization benchmarks like DomainBed (Gulrajani & Lopez-Paz, 2021a). In other words, we train on all domains but one, compute accuracy on the test sets for both ID and OOD data, and average the results. Robutness to label noise is measured using real-world noise (*i.e.*, noise that appears naturally) as well as controlled experiments like those we conducted using RotatedMNIST in Section 3 and Section 4. Real-world noise experiments validate that our approach works in practice, while the controlled experiments enable us to provide into deeper insights by controlling the noise that appears in the datasets.

Following Ballas & Diou (2025); Gulrajani & Lopez-Paz (2021a); Wang et al. (2025), our experiments use a ResNet-50 (He et al., 2016) pretrained on ImageNet (Deng et al., 2009), except for CHAMMI-CP which, following Chen et al. (2024b), uses a ConvNeXt (Liu et al., 2022) pretrained on ImageNet 22K. While some prior work has used large multimodal models (LMMs) in their experiments, LMMs have applications restrictions due to task or computational requirements. *E.g.*, CHAMMI-CP (Chen et al., 2024b) has up to 5 channel cell images, so LMMs that process 3 channel natural images do not apply. Additionally, recent work has shown the high performance of LMMs

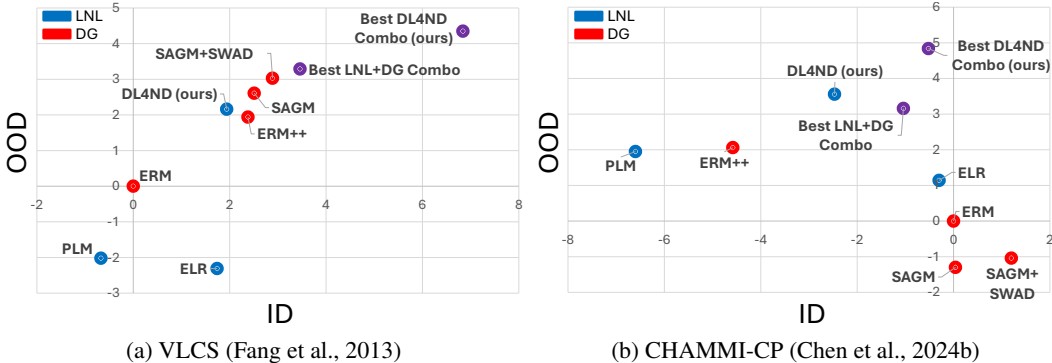

(a) VLCS (Fang et al., 2013)  (b) CHAMMI-CP (Chen et al., 2024b)

Figure 6: Summary of real-world noise top-1 accuracy changes on VLCS (Fang et al., 2013) and CHAMMI-CP (Chen et al., 2024b). We center results around ERM (Gulrajani & Lopez-Paz, 2021a), and report relative absolute changes in both in-domain (ID) and out-of-domain (OOD) performance for the best LNL (ELR (Liu et al., 2020), PLM (Zhao et al., 2024)) and DG methods (ERM++ (Teterwak et al., 2025b), SAGM (Wang et al., 2023), SWAD (Cha et al., 2021)). Appendix A contains additional methods and combinations. We find DL4ND outperforms prior LNL methods, even when combined with DG approaches. See Section 5.1 for further discussion.

Table 2: Real-world noise top-1 accuracy on PACS (Li et al., 2017a). Integrating DL4ND with DG methods provides up to a 2% average gain.

| Method | ID | OOD | AVG |
|---|---|---|---|
| SAGM | 96.3 | 85.3 | 90.8 |
| w/ DL4ND (ours) | **97.3** | 88.8 | 93.1 |
| ERM++ | 96.7 | 89.2 | 92.9 |
| w/ DL4ND (ours) | 96.5 | **90.1** | **93.3** |

Table 3: Results with 60% symmetric noise on Office-Home (Venkateswara et al., 2017) (see Table 4 for asymmetric noise). Using DL4ND provides up to 12.5% gain.

| Method | ID | OOD | AVG |
|---|---|---|---|
| ERM | 45.8 | 40.5 | 43.2 |
| w/ DL4ND (ours) | 47.9 | 49.9 | 48.9 |
| SAGM | 48.6 | 40.3 | 44.4 |
| w/ DL4ND (ours) | 52.0 | 52.6 | 52.2 |
| ERM++ | 56.7 | 48.7 | 52.7 |
| w/ DL4ND (ours) | **60.3** | **59.4** | **59.8** |

on DG benchmarks like VLCS and OfficeHome is largely due to train/test contamination (Teterwak et al., 2025a), which is significantly less pronounced in TerraIncognita. This also likely explains why some of our baseline methods outperform LLMs like GPT-4v (by up to 15%) reported in Han et al. (2024) on TerraIncognita using a ResNet-50. Additional implementation details are in Appendix D.

## 5.1 RESULTS

Figure 6 summarizes key results comparing our DL4ND approach with top methods from the LNL and DG literature, along with their best combinations, on VLCS (Fang et al., 2013) and CHAMMI-CP (Chen et al., 2024b) (additional results in Appendix A). We make two notable observations. First, our DL4ND method on its own outperforms other LNL methods, and DL4ND is the only approach on CHAMMI-CP that provides an average boost on its own. Second, DL4ND combined with DG methods outperforms combinations of prior work in DG+LNL by 1-2% on average. Table 2 demonstrates that the advantage from DL4ND also extends to PACS (Li et al., 2017a), booting performance when combined with DG methods by up to 2%. These results show that DL4ND is beneficial across diverse datasets with real-world noise, especially when combined with DG methods.

Table 3 and Table 4 provides results on a controlled experiments where we add symmetric noise to OfficeHome (Venkateswara et al., 2017) and asymmetric noise to OfficeHome, TerraIncognita (Beery et al., 2018), and DomainNet (Peng et al., 2019), respectively. Overall, DL4ND results in a boost to performance in most settings, obtaining the best performance in 11 of 13 cases, with gains as large as 12.5% (OOD results on symmetric OfficeHome). We find that as we increase the amount of noise in a dataset that the benefit of DL4ND also grows.

Table 4: Asymmetric noise results on OfficeHome (Venkateswara et al., 2017), TerraIncognita (Beery et al., 2018), and DomainNet (Peng et al., 2019). We **bold** overall best results, and underline best results between DL4ND and a specific DG method it is integrated with. DL4ND boosts performance in 8 of 10 settings, with a 9% gain on DomainNet. More results are in Table 11.

| | **OfficeHome** | | | | | | **TerraIncognita** | | | | | | **DomainNet** | |
|---|---|---|---|---|---|---|---|---|---|---|---|---|---|---|
| Method | No Noise | | 20% Noise | | 40% Noise | | No Noise | | 20% Noise | | 40% Noise | | 40% Noise | |
| | ID | OOD | ID | OOD | ID | OOD | ID | OOD | ID | OOD | ID | OOD | ID | OOD |
| SAGM | 83.4 | 69.1 | 76.7 | 64.0 | 62.0 | 52.0 | 86.7 | 51.3 | **78.2** | 40.4 | 56.4 | 30.9 | 43.6 | 26.9 |
| w/ DL4ND (ours) | – | – | **81.4** | 66.6 | 69.3 | 55.0 | – | – | 77.1 | 43.5 | **58.7** | 32.8 | **52.8** | 35.6 |
| ERM++ | 85.1 | 71.7 | 78.5 | 65.8 | 63.1 | 52.3 | 85.9 | 47.5 | 75.9 | **45.0** | 56.1 | 30.6 | 42.4 | 28.9 |
| w/ DL4ND (ours) | – | – | 76.6 | **68.6** | 62.7 | **56.4** | – | – | 73.0 | 43.2 | 57.0 | **37.5** | 51.1 | **36.2** |

Table 5: Ablation study on VLCS (Fang et al., 2013) and CHAMMI-CP (Chen et al., 2024b). We **bold** overall best results, and underline best results between when DL4ND is integrated and its ablations. We show our model components boost performance by 2-4%.

| Method | VLCS | | | CHAMMI-CP | | |
|---|---|---|---|---|---|---|
| | ID | OOD | AVG | ID | OOD | AVG |
| SAGM+SWAD w/DL4ND (ours) | 91.9 | 88.6 | 90.6 | **76.6** | 47.3 | **61.9** |
| w/o relabel | 88.5 | 87.2 | 87.8 | 70.9 | 47.2 | 59.0 |
| w/o cross-domain | 88.6 | 87.8 | 88.2 | 68.5 | 49.0 | 58.8 |
| w/o small-loss proxy | 88.2 | 87.0 | 87.6 | 73.6 | 45.8 | 59.7 |
| ERM++ w/DL4ND (ours) | **95.3** | **89.0** | **92.2** | 72.9 | 44.3 | 58.6 |
| w/o relabel | 89.5 | 88.8 | 89.2 | 72.5 | 45.4 | 58.9 |
| w/o cross-domain | 89.5 | 87.6 | 88.6 | 69.7 | 47.3 | 58.5 |
| w/o small-loss proxy | 93.5 | 87.7 | 90.6 | 68.9 | 43.4 | 56.1 |

Table 5 provides an ablation study that shows each component of DL4ND improves performance. Specifically, *w/o relabel* refers to removing rather than relabeling samples, *w/o cross-domain* uses within-domain rather than cross-domain comparisons to identify noise (described in Section 4.1), and *w/o small-loss proxy* reports results that uses all samples to create the proxies for cross-domain comparisons rather than just low-loss samples (discussed in Section 3.1). As shown in Table 5, each component provides a 2-4% gain in most settings, with the best performing method on each dataset using all components. In Table 6 we also show that the gain from cross-domain comparisons can at least be partly explained as an improvement in the precision of relabeled samples.

## 5.2 WHY DON'T NAIVE LNL+DG COMBINATIONS PERFORM BETTER?

At a high level, DL4ND resembles UNICON (Karim et al., 2022), which also estimates and relabels potential noisy samples. However, as discussed in the Introduction, LNL methods like UNICON assume that high-loss samples tend to be noise, but in NAG these samples may also be domains that are harder to learn. For example, Figure 7 shows that samples selected as part of the clean subset when training on VLCS mostly come from LabelMe, whereas some domains like SUN09 are mostly identified as noise. This makes it challenging to learn from all domains as some may have many samples incorrectly relabeled (especially those along the decision boundary). In effect, this relabeling introduces more noise rather than fixing it. Figure 8 in the Appendix shows these issues persist even when decreasing ratio of samples allowed to be relabeled. This observation is similar to the one that UNICON made of DivideMix (Li et al., 2020), where they noted that some categories were more prone to relabeling than others and argued for more balanced approach to address it.

In Table 7 we extend this idea to the multi-domain setting, where we ensure relabeling is also source-balanced. Specifically, we perform per-domain sampling, where we relabel only a subset of each domain (*i.e.*, we relabel 20% of the samples in each domain rather than 20% of the dataset regard-

Table 6: Comparing relabeling precision at two noise levels on OfficeHome (Venkateswara et al., 2017) and TerraIncognita (Beery et al., 2018). Our cross-domain comparisons improves precision by up to 10% over using a single domain.

| Comparison | OfficeHome | | TerraIncognita | |
|---|---|---|---|---|
| | 20% | 40% | 20% | 40% |
| same-domain | 99.5 | 94.0 | 82.5 | 64.3 |
| cross-domain | **99.8** | **97.2** | **86.7** | **74.3** |

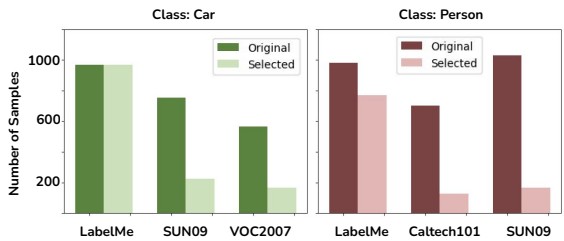

Figure 7: **Changes in domain distribution after the UNICON sample selection process on VLCS** (Fang et al., 2013). (*Left bar: number of samples before selection, right bar: after selection*). These two cases illustrate a risk of skewing domain distributions from the LNL selection process. See Sec. 5.2 for more details.

Table 7: Comparison of relabeling methods on real-world noise top-1 accuracy on VLCS (Fang et al., 2013) and CHAMMI-CP (Chen et al., 2024b). We use the best performing DG method SAGM (Wang et al., 2023)+SWAD (Cha et al., 2021) and combine them with to relabeling method UNICON (Karim et al., 2022) or DL4ND. *Per-domain sampling* uses domain labels to separate samples by their sources, and relabels samples evenly across domains. This effectively reduces to the standard LNL task addressed by UNICON, since each source does not contain domain shifts. However, DL4ND still performs better, especially on VLCS. See Section 5.2 for discussion.

| DG Method | LNL Method | Per-domain | VLCS | | | CHAMMI-CP | | |
|---|---|---|---|---|---|---|---|---|
| | | sampling | ID | OOD | AVG | ID | OOD | AVG |
| – | UNICON | – | 89.9 | 84.0 | 86.9 | 76.7 | 42.0 | 59.4 |
| SAGM+SWAD | – | – | 91.4 | 87.7 | 89.5 | **78.3** | 41.5 | 59.9 |
| SAGM+SWAD | UNICON | – | 87.8 | 82.8 | 85.3 | 75.4 | 43.3 | 59.3 |
| SAGM+SWAD | UNICON | ✓ | 89.3 | 86.8 | 88.1 | 77.0 | 46.3 | 61.6 |
| SAGM+SWAD | DL4ND (ours) | – | **91.9** | **88.6** | **90.3** | 76.6 | **47.3** | **61.9** |

less of domain). Table 7 shows this does improve the performance of combining UNICON with DG methods SAGM+SWAD by 2-3% on average. However, it only improves performance compared to SAGM+SWAD alone on CHAMMI-CP, and actually decreases performance on VLCS. This is likely due, in part, to the fact that in this setting UNICON will still utilize same-domain comparisons, which we showed in the previous section underperforms DL4ND's cross-domain approach. Thus, as seen comparing the last two lines of Table 7, our approach is 2% better on VLCS while also providing a small gain on CHAMMI-CP. Notably, DL4ND improves performance over SAGM+SWAD alone on both datasets unlike UNICON.

## 6 CONCLUSION

This work addresses the challenges of training noisy, diverse real-world data by exploring Noise-Aware Generalization (NAG), which focuses on handling in-domain noise and improving out-of-domain generalization. We highlight several key takeaways. First, NAG presents new challenges, which complicate the task of distinguishing between noise and domain distribution shifts. Second, a naive combination of LNL and DG does not effectively address this task. Domain shift can interfere with noise detection, and LNL-based sample selection can inadvertently skew the domain distribution. Lastly, we demonstrate that using cross-domain comparisons as a critical signal for noise detection significantly improves performance. Noise, which lacks the intrinsic class features, fails to exhibit closer distances to other domains. Experimental results validate the effectiveness of our approach, and the discussion also provides insights for further advancing NAG.

**Acknowledgments.** This study was supported by the National Science Foundation under NSF-DBI award 2134696. Any opinions, findings, and conclusions or recommendations expressed in this material are those of the author(s) and do not necessarily reflect the views of the supporting agency.

## ETHICS STATEMENT

This paper addresses NAG, a task that requires models to perform well on both in-domain and out-of-domain data when training on datasets with label noise. This can result in models that can effectively learn from a wide variety of data, including cell painting data where prior work in tasks like LNL found especially challenging due to its high amounts of label noise (Wang & Plummer, 2024). However, like many topics in this field, also can enable bad actors to use these models to train more effective recognition systems for nefarious purposes. Additionally, users should be mindful that although we provide an evaluation on a diverse set of datasets, they still make mistakes in their predictions that may vary depending on the dataset. Thus, researchers and engineers should be mindful of these factors when deploying a system for end-users.

## REPRODUCIBILITY STATEMENT

We have released our code and data to ensure it can be reproduced. We provide detailed implementation details as well as an in-depth example of how to integrate the methods in our experiments in Appendix D. Our released code is capable of training and testing the models we compared to in a unified codebase. This enables additional methods to be easily integrated and the data loaders required to evaluate models on our benchmarks. Further, we have also included some pretrained models for ease of use.

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

Table 8: Comparing methods of addressing NAG on VLCS (Fang et al., 2013) and CHAMMI-CP (Chen et al., 2024b) either via (a) by using a method from prior work in the DG literature or (b) LNL methods like our approach DL4ND approach. We bold the best performance overall and underline the best LNL method. We find DL4ND performs best among LNL methods, and does especially well on CHAMMI-CP. See Appendix A for discussion.

|  | Method | VLCS | | | CHAMMI-CP | | |
|---|---|---|---|---|---|---|---|
|  |  | ID | OOD | AVG | ID | OOD | AVG |
| (a) | ERM (Gulrajani & Lopez-Paz, 2021a) | 88.5 | 84.6 | 86.6 | 77.1 | 42.5 | 59.8 |
|  | VREx (Krueger et al., 2021) | 89.0 | 84.4 | 86.7 | 74.8 | 44.8 | 59.8 |
|  | SWAD (Cha et al., 2021) | 90.8 | 86.2 | 88.5 | 73.9 | 43.7 | 58.8 |
|  | Fishr (Rame et al., 2022) | 88.9 | 85.8 | 87.4 | 73.9 | 44.0 | 59.0 |
|  | MIRO (Cha et al., 2022) | 88.9 | 83.8 | 86.4 | 65.5 | **46.6** | 56.0 |
|  | SAGM (Wang et al., 2023) | 91.0 | 87.2 | 89.1 | 77.1 | 41.2 | 59.2 |
|  | DISAM Zhang et al. (2024) | 89.6 | 84.9 | 87.2 | 72.4 | 44.8 | 58.6 |
|  | ERM++ (Teterwak et al., 2025b) | 90.9 | 86.6 | 88.7 | 72.5 | 44.6 | 58.5 |
|  | MIRO+SWAD | 88.9 | 83.7 | 86.3 | 67.3 | 45.8 | 56.6 |
|  | SAGM+SWAD | **91.4** | **87.7** | **89.5** | **78.3** | 41.5 | 59.9 |
| (b) | ELR (Liu et al., 2020) | 90.3 | 82.3 | 86.3 | 76.8 | 43.6 | 60.2 |
|  | UNICON (Karim et al., 2022) | 89.9 | 84.0 | 86.9 | 76.7 | 42.0 | 59.4 |
|  | DISC (Li et al., 2023) | 88.7 | 82.5 | 85.6 | 43.3 | 41.3 | 42.3 |
|  | PLM (Zhao et al., 2024) | 87.9 | 82.6 | 85.2 | 70.5 | 44.4 | 57.5 |
|  | DL4ND (ours) | 90.5 | 86.8 | 88.6 | 74.6 | 46.0 | **60.3** |

Yivan Zhang, Gang Niu, and Masashi Sugiyama. Learning noise transition matrix from only noisy labels via total variation regularization. In *International Conference on Machine Learning*, pp. 12501–12512. PMLR, 2021.

Rui Zhao, Bin Shi, Jianfei Ruan, Tianze Pan, and Bo Dong. Estimating noisy class posterior with part-level labels for noisy label learning. In *Proceedings of the IEEE/CVF Conference on Computer Vision and Pattern Recognition*, pp. 22809–22819, 2024.

Junbao Zhuo, Shuhui Wang, and Qingming Huang. Uncertainty modeling for robust domain adaptation under noisy environments. *Trans. Multi.*, pp. 6157–6170, January 2023.

Yukun Zuo, Hantao Yao, Liansheng Zhuang, and Changsheng Xu. Seek common ground while reserving differences: A model-agnostic module for noisy domain adaptation. *IEEE Transactions on Multimedia*, 24:1020–1030, 2022. doi: 10.1109/TMM.2021.3097495.

# A  EXTENDED RESULTS

Table 8 contains detailed results for both DG and LNL-only methods. Generally speaking, the results follow intuition, where DG methods perform better on VLCS, which has less noise and more dramatic domain shifts. In contrast, DL4ND being an LNL method performs the best on CHAMMI-CP, which has more noise and smaller shifts. That said, our approach does outperform other LNL methods on VLCS by 2%. Notably, for DG methods the more regularization focused methods like SWAD are necessary to get best performance. This observation is key as in the LNL methods ELR is also a regularization-style approach and performs similarly to DL4ND on CHAMMI-CP. Thus, conceptually our approach has more differences than ELR compared to the DG methods that do well on our NAG task.

As shown in Table 9, this conceptual difference manifests itself when we aim to combine DG and LNL methods. Notably, while ELR actually performs worse on CHAMMI-CP, suggesting that the model becomes too constrained with the additionl of multiple types of regularization, DL4ND performs better, getting a 2% boost on CHAMMI and a 3.5% gain on VLCS when combined with DG method (comparing the last line of Table 8 to the last two lines of Table 9). The effectiveness of

Table 9: Comparing different combinations of DG methods on VLCS (Fang et al., 2013) and CHAMMI-CP (Chen et al., 2024b) with (a) LNL methods from prior work or (b) our DL4ND approach. We find combinations with our DL4ND method gets best performance. See Appendix A for discussion.

| | DG Method | LNL Method | VLCS | | | CHAMMI-CP | | |
|---|---|---|---|---|---|---|---|---|
| | | | ID | OOD | AVG | ID | OOD | AVG |
| (a) | ERM++ | ELR | 89.7 | 85.4 | 87.6 | 75.7 | 42.0 | 58.9 |
| | MIRO+SWAD | ELR | 91.5 | 86.7 | 89.1 | 70.7 | 44.8 | 57.8 |
| | MIRO | ELR | 90.8 | 84.5 | 87.7 | 74.5 | 41.3 | 57.9 |
| | SWAD | ELR | 92.0 | 87.9 | 90.0 | 73.5 | 44.7 | 59.1 |
| | MIRO | UNICON | 89.8 | 83.4 | 86.6 | **77.0** | 43.4 | 60.2 |
| | MIRO+SWAD | UNICON | 88.9 | 83.7 | 86.3 | 76.0 | 45.7 | 60.8 |
| | SAGM+SWAD | UNICON | 87.8 | 82.8 | 85.3 | 75.4 | 43.3 | 59.3 |
| (b) | VREx | DL4ND (ours) | 91.2 | 87.0 | 89.1 | 75.3 | 46.7 | 61.0 |
| | Fishr | DL4ND (ours) | 89.9 | 86.5 | 88.2 | 73.8 | 46.1 | 60.0 |
| | MIRO | DL4ND (ours) | 93.5 | 86.7 | 90.1 | 70.4 | 46.7 | 58.5 |
| | MIRO+SWAD | DL4ND (ours) | 91.7 | 88.0 | 89.9 | 71.2 | 46.6 | 58.9 |
| | SAGM | DL4ND (ours) | 91.9 | 88.4 | 90.1 | 76.2 | 46.6 | 61.4 |
| | SAGM+SWAD | DL4ND (ours) | 91.9 | 88.6 | 90.3 | 76.6 | **47.3** | **61.9** |
| | ERM++ | DL4ND (ours) | **95.4** | **89.0** | **92.2** | 72.9 | 44.3 | 58.6 |

Table 10: Comparison of relabeling methods on real-world noise top-1 accuracy on VLCS (Fang et al., 2013) and CHAMMI-CP (Chen et al., 2024b) expanding on our results from Section 5.2. *Per-domain sampling* uses domain labels to separate samples by their sources, and relabels samples evenly across domains. DL4ND outperforms UNICON in nearly all cases.

| DG Method | LNL Method | Per-domain sampling | VLCS | | | CHAMMI-CP | | |
|---|---|---|---|---|---|---|---|---|
| | | | ID | OOD | AVG | ID | OOD | AVG |
| MIRO | UNICON | ✓ | 91.2 | 85.8 | 88.5 | 76.9 | 45.2 | 61.1 |
| MIRO+SWAD | UNICON | ✓ | 90.6 | 86.0 | 88.3 | 76.5 | 43.6 | 60.0 |
| SAGM+SWAD | UNICON | ✓ | 89.3 | 86.8 | 88.1 | **77.0** | 46.3 | 61.6 |
| SAGM+SWAD | DL4ND (ours) | – | **91.9** | **88.6** | **90.3** | 76.6 | **47.3** | **61.9** |

the DG methods also changes when combined with DL4ND, as SAGM+SWAD performed best in Table 8 on VLCS, but we get nearly a 3% gain combining ERM++ with DL4ND in Table 9.

In addition, as we discussed in Section 5.2, part of the reason why UNICON does not perform as well as DL4ND in Table 9 is due to the fact that it tends to be biased towards detecting samples from a subset of domains as noise. *I.e.*, harder to learn domains incorrectly get identified as largely noise. Table 10 expands on our results from the main paper where we assume we are provided with domain labels, which enables us to per-domain sampling. On VLCS all the UNICON results underperforms DL4ND by 2%. The various combinations of UNICON with DG methods generally perform the same (even though they do all outperform combinations in Table 9), with the only real variation combing on CHAMMI-CP. However, even on this dataset our DL4ND reports a slight advantage, mostly due to stronger OOD performance.

Table 11 reports an ERM baseline (Gulrajani & Lopez-Paz, 2021a) on its own and when combined with our approach on OfficeHome (Venkateswara et al., 2017) and TerraIncognita (Beery et al., 2018) to supplement the results in the main paper. We find that DL4ND improves performance by 3-18%, demonstrating that our approach provide a significant benefit over DG methods alone.

Table 12 and Table 13 provides a sensitivity analysis on warmup length on PACS (Li et al., 2017a) and TerraIncognita (Beery et al., 2018), respectively. Notably, the warmup length on either dataset is relatively short, but should be aided by a validation set to optimize its length. That said, as the training dynamics during warmup does not change, setting the warmup length can be done in a single

Table 11: Synthetic noise results on OfficeHome (Venkateswara et al., 2017) and TerraIncognita (Beery et al., 2018) to supplement results from Section 5.1. DL4ND boosts performance by 3-18% over ERM (Gulrajani & Lopez-Paz, 2021a) alone.

| Method | OfficeHome | | | | | | TerraIncognita | | | |
| | No Noise | | 20% Noise | | 40% Noise | | No Noise | | 40% Noise | |
| | ID | OOD | ID | OOD | ID | OOD | ID | OOD | ID | OOD |
|---|---|---|---|---|---|---|---|---|---|---|
| ERM | 80.6 | 65.6 | 71.9 | 59.6 | 57.8 | 46.6 | 84.1 | 46.3 | 53.0 | 33.8 |
| w/ DL4ND (ours) | – | – | **80.2** | **64.7** | **68.4** | **54.1** | – | – | **56.3** | **37.2** |

Table 12: Sensitivity analysis to warmup length on real-world noise top-1 accuracy on PACS (Li et al., 2017a). See Appendix A for discussion.

| Warmup Length | ID | OOD | AVG |
|---|---|---|---|
| w/o DL4ND | 96.3 | 85.3 | 90.8 |
| 300 steps | 96.7 | 84.9 | 90.8 |
| 600 steps | **97.3** | **88.8** | **93.1** |
| 1000 steps | 97.2 | 88.6 | 92.9 |

run (*i.e.*, using a validation set to identify when performance gains start to decelerate). In addition, performance on PACS especially shows that the exact length choice can be relatively insensitive.

### A.1 CHANGING SAMPLE SELECTION RATIOS

Figure 8 shows the relationship between domain balance, clean sample count, and ID/OOD performance for the "person" class in VLCS. At lower selection ratios ($r$), the selected samples are cleaner but the distribution skews toward the cleaner VOC2007 domain, while higher ratios maintain balance but increase noise. The best results occur at $r = 0.2$, indicating that quality outweighs quantity for improved robustness.

## B COMPARISON TO DA+LNL METHODS

As discussed in our related work section, where are several methods that explore the intersection of Domain Adaption (DA) and LNL (*e.g.*, Shu et al. (2019); Han et al. (2023); Zuo et al. (2022); Zhuo et al. (2023); Feng et al. (2023); Yin et al. (2025)). In DA their goal is to improve performance on a target domain when given a labeled source domain and unlabeled samples from a target domain. However, in DG, which we study, the goal is to generalize to unseen distributions, thereby requiring different methods. Additionally, they are evaluated on how well they perform on the target domain, which is similar to DG methods that evaluate performance only on unseen domains (ignoring source domain performance). As we show, this often results in performing well only on the unseen domains, thereby limiting its applications when both ID and OOD performance is desired (*e.g.*, MIRO in Table 9(a) obtains a 4% boost to OOD performance on CHAMMI, but at a cost to 12% ID performance).

Further, when we consider adapting the ideas of these methods to our task, we find that many of them are well represented in the LNL literature. For example, Yin et al. (2025); Zuo et al. (2022)

Table 13: Sensitivity analysis to warmup length on 20% asymmetric noise on TerraIncognita (Beery et al., 2018). See Appendix A for discussion.

| Warmup Length | ID | OOD | AVG |
|---|---|---|---|
| w/o DL4ND | **78.2** | 40.6 | 59.3 |
| 1000 steps | 77.4 | 41.2 | 59.3 |
| 1200 steps | 77.1 | **43.5** | **60.3** |
| 1500 steps | 77.1 | 41.4 | 59.3 |

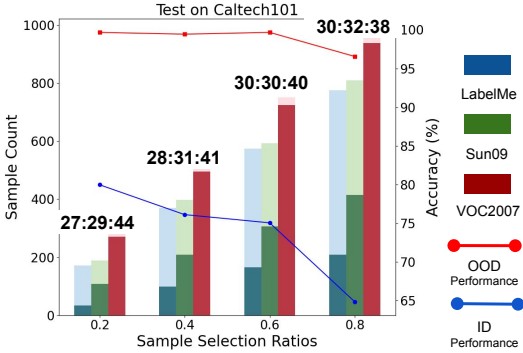

Figure 8: **Balance, clean sample ratios, and ID/OOD performance on VLCS "Person" class.** Testing on Caltech101 with training on other domains. The x-axis shows sample selection ratios per class, with domain ratios above the bars. (*Dark: clean samples; light: noisy.*) The decline in ID and OOD performance as balance increases suggests that a more balanced distribution does not always improve OOD accuracy, and increased noise harms both ID and OOD. See Appendix A.1 for discussion.

separates samples into a clean and noisy set, and then aims to use the clean set to correct the noisy labels, which is also the general approach taken by relabeling methods like DISC (Li et al., 2023) and UNICON (Karim et al., 2022). Similarly, Feng et al. (2023); Zhuo et al. (2023) aims to estimate label uncertainty and then re-weight samples based on this uncertainty, which is also the general approach of LNL method PLM (Zhao et al., 2024). As seen in Table 9(b), DISC, UNICON, and PLM underperform our DL4ND approach by up to 18%.

The reason these methods do not transfer well is due, at least in part, to the differences in DA and DG, as there tends to only be a transfer between a single source domain and a single target domain. Specifically, if we consider the example in Figure 1 of our paper, the main problem we raise is that in NAG we must learn to separate label noise from domain shifts. However, in DA there is no need to solve this problem as there is only a single source domain with noise. Thus, they are more akin to the traditional LNL task that aims to identify noise in a single domain, which is likely why similar ideas appear between them. We simulate this type of setting by using per-domain sampling in Table 7, effectively converting the problem into a set of single-domain noise detection tasks. However, this underperforms our approach by 2% on VLCS. This not only highlights the differences between the tasks but also illustrates the benefits of our cross-domain comparisons.

## C   REAL-WORLD DATASET'S NOISE

Figure 9 reports statistics on the two datasets used most extensively in our experiments, VLCS (Fang et al., 2013) and CHAMMI-CP (Chen et al., 2024b). Generally speaking, CHAMMI-CP is larger, with more domains and is very noisy. However, while there are many domains, these represent what can be thought of as reproduction experiments. *I.e.*, the variations that are shown are due to technical variations that arise when reproducing an experiment within the same laboratory setting (*i.e.*, same equipment). As such, their domains are more similar than those in VLCS.

Table 14 provides more detailed statistics about the source of the noise in VLCS. Notably, the various domains have different degrees of noise, *e.g.*, Caltech (Fei-Fei et al., 2004) is relatively clean, whereas LabelMe (Russell et al., 2008) and SUN09 (Choi et al., 2010) are much noisier. The main source of the noise is due to unlabeled categories. For example, person images that do not label cars or chairs are common. As such, while it is the person images being labeled as noisy, the actual issue is that car is not also annotated. This makes this noise type similar to the asymmetric noise used in our synthetic noise experiments in Section 5. Our inspection of PACS (Li et al., 2017a) finds it contains the same type of noise as VLCS.

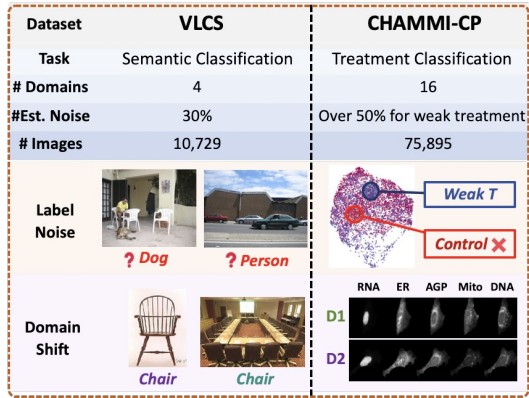

Figure 9: **Real-world datasets with in-domain noise and multi-domain distribution**. **VLCS** (web/user data) (Fang et al., 2013), and **CHAMMI-CP** (biomedical images) (Chen et al., 2024b). VLCS faces label noise from poor annotations and domain shifts from varying data sources, while CHAMMI-CP deals with ambiguous features and varying experimental environments.

## C.1 SYNTHESIZED NOISE

Our synthetic experiments use asymmetric noise, which flips labels between commonly confused categories. These confusions can, but are not required to go both directions, and are manually selected. We provide our category confusion information in Table 15 for RotatedMNIST (Ghifary et al., 2015), Table 16 for OfficeHome (Venkateswara et al., 2017), Table 17 for TerraIncognita (Beery et al., 2018), and Table 18 for DomainNet (Peng et al., 2019).

## D IMPELMENTATION DETAILS

We incorporate the implementation of the ERM++ [2] (Teterwak et al., 2025b), DISC [3] (Li et al., 2023), UNICON [4] (Karim et al., 2022), ELR [5] (Liu et al., 2020), SAGM [6] (Wang et al., 2023), MIRO [7] (Cha et al., 2022), VREx [8] (Krueger et al., 2021), Fishr [9] (Rame et al., 2022), DISAM [10] (Zhang et al., 2024), PLM [11] (Zhao et al., 2024), into our codebase. Each training batch includes samples from all training domains, with a batch size of 128. For relatively small datasets VLCS (Fang et al., 2013) and CHAMMI-CP (Chen et al., 2024b), experiments are run on a single NVIDIA RTX A6000 (48GB RAM) and three Intel(R) Xeon(R) Gold 6226R CPU @ 2.90GHz for 5000 steps. Below we give a short description of the methods in prior work used in our experiments.

## D.1 LEARNING WITH NOISY LABELS METHODS

**ELR** (Liu et al., 2020) is based on the observation that deep neural networks initially fit the training data with clean labels during an "early learning" phase before eventually memorizing examples with false labels. It employs semi-supervised learning to generate target probabilities and introduces a regularization term to prevent memorizing false labels, guiding the model toward these target probabilities.

---

[2]https://github.com/piotr-teterwak/erm_plusplus

[3]https://github.com/JackYFL/DISC

[4]https://github.com/nazmul-karim170/UNICON-Noisy-Label

[5]https://github.com/shengliu66/ELR

[6]https://github.com/Wang-pengfei/SAGM

[7]https://github.com/kakaobrain/miro

[8]https://github.com/facebookresearch/DomainBed

[9]https://github.com/alexrame/fishr

[10]https://github.com/MediaBrain-SJTU/DISAM

[11]https://github.com/RyanZhaoIc/PLM/tree/main

Table 14: VLCS Dataset Overview (Total Samples, Noisy Samples) over its datasets Caltech (Fei-Fei et al., 2004), LabelMe (Russell et al., 2008), SUN09 (Choi et al., 2010), and VOC2007 (Everingham et al., 2010).

| Domain | Category | Total Samples | Noisy Samples |
|---|---|---|---|
| Caltech | Bird | 237 | 1 (with person) |
| | Car | 123 | 0 (black & white car imgs) |
| | Chair | 118 | 0 |
| | Dog | 67 | 0 (only black and white dog) |
| | Person | 870 | 0 (profile photos with redundancy) |
| LabelMe | Bird | 80 | 20 |
| | Car | 1209 | 559 (background: building, road, mountains; small & incomplete cars, unclear night imgs [OOD]) |
| | Chair | 89 | 61 (over half have cars, person) |
| | Dog | 43 | 25 (with person, cars) |
| | Person | 1238 | 924 (over 80% noisy images have cars, street photos are similar to car and chair categories, small person figures) |
| SUN09 | Bird | 21 | 12 (background, 1 person and dog) |
| | Car | 933 | 548 (street view, buildings, person) |
| | Chair | 1036 | 186 (mostly person, very few car interior) |
| | Dog | 31 | 25 ($\sim$20 noisy images with person) |
| | Person | 1265 | 631 (very small person figures) |
| VOC2007 | Bird | 330 | 29 (mostly human, a few cars, one small bird) |
| | Car | 699 | 133 (mostly person, $\sim$5 don't have cars) |
| | Chair | 428 | 145 (mostly person, some cars, very few missing chair) |
| | Dog | 420 | 111 (mostly human, a few cars) |
| | Person | 1499 | 61 (mostly cars, some don't have person) |

**UNICON** (Karim et al., 2022) partitions the training set into clean and noisy subsets using uniform selection at each iteration. It estimates the clean label probability via Jensen-Shannon divergence (JSD) based on prediction and one-hot label distribution disagreement. After partitioning, UNICON applies semi-supervised learning (SSL) with a contrastive loss across two identical networks, repeating this process until convergence.

**DISC** (Li et al., 2023) identifies clean samples based on predictions from weak and strong augmentations, using a dynamic confidence threshold determined by each instance's memorization strength from previous epochs. This approach classifies instances into clean, hard, and purified subsets. Different regularization strategies are then applied to each subgroup.

Table 15: Asymmetric Noise Pairs for RotatedMNIST (Ghifary et al., 2015).

| Index A | Index B |
|---------|---------|
| 0 | 6 |
| 1 | 7 |
| 3 | 5 |
| 4 | 9 |
| 5 | 3 |
| 6 | 0 |
| 7 | 1 |
| 9 | 4 |

Table 16: Asymmetric Noise Pairs for OfficeHome (Venkateswara et al., 2017).

| Index A | Class A | Index B | Class B |
|---------|---------|---------|---------|
| 16 | Pencil | 6 | Pen |
| 14 | Keyboard | 42 | Laptop |
| 15 | Mouse | 60 | Monitor |
| 10 | Backpack | 39 | Clipboards |
| 1 | Calculator | 34 | Notebook |
| 47 | Bottle | 63 | Soda |
| 13 | Flowers | 21 | Candles |
| 3 | Flipflops | 54 | Sneakers |
| 9 | TV | 60 | Monitor |
| 8 | Speaker | 53 | Radio |
| 4 | Kettle | 52 | Pan |
| 19 | Webcam | 42 | Laptop |
| 5 | Mop | 56 | Bucket |
| 24 | Knives | 32 | Fork |
| 12 | Desk Lamp | 33 | Lamp Shade |
| 18 | Spoon | 32 | Fork |
| 17 | Scissors | 27 | Screwdriver |
| 50 | Hammer | 22 | Drill |
| 48 | Computer | 60 | Monitor |
| 23 | Folder | 34 | Notebook |
| 26 | Post-it Notes | 61 | Paper Clip |
| 58 | File Cabinet | 36 | Shelf |
| 44 | Push Pin | 26 | Post-it Notes |
| 45 | Sink | 62 | Refrigerator |
| 49 | Fan | 33 | Lamp Shade |
| 25 | Mug | 47 | Bottle |
| 57 | Couch | 30 | Chair |

Table 17: Asymmetric Noise Pairs for TerraIncognita (Beery et al., 2018).

| Index A | Class A | Index B | Class B |
|---------|---------|---------|---------|
| 0 | Bird | 9 | Squirrel |
| 1 | Bobcat | 3 | Coyote |
| 2 | Cat | 4 | Dog |
| 3 | Coyote | 8 | Raccoon |
| 4 | Dog | 2 | Cat |
| 5 | Empty | 0 | Bird |
| 6 | Opossum | 8 | Raccoon |
| 7 | Rabbit | 9 | Squirrel |
| 8 | Raccoon | 6 | Opossum |
| 9 | Squirrel | 7 | Rabbit |

Table 18: Asymmetric Noise Pairs for DomainNet (Peng et al., 2019).

| Key | Value | Key | Value | Key | Value | Key | Value | Key | Value | Key | Value |
|---|---|---|---|---|---|---|---|---|---|---|---|
| 0 | 308 | 1 | 208 | 2 | 28 | 3 | 135 | 4 | 5 | 5 | 0 |
| 6 | 0 | 7 | 324 | 8 | 324 | 9 | 208 | 10 | 288 | 11 | 324 |
| 12 | 208 | 13 | 285 | 14 | 208 | 15 | 16 | 16 | 17 | 17 | 282 |
| 18 | 19 | 19 | 327 | 20 | 309 | 21 | 208 | 22 | 327 | 23 | 208 |
| 24 | 288 | 25 | 135 | 26 | 27 | 27 | 28 | 28 | 208 | 29 | 208 |
| 30 | 327 | 31 | 98 | 32 | 33 | 33 | 144 | 34 | 35 | 35 | 308 |
| 36 | 282 | 37 | 38 | 38 | 327 | 39 | 208 | 40 | 208 | 41 | 42 |
| 42 | 208 | 43 | 44 | 44 | 308 | 45 | 46 | 46 | 331 | 47 | 324 |
| 48 | 91 | 49 | 90 | 50 | 327 | 51 | 324 | 52 | 53 | 53 | 324 |
| 54 | 327 | 55 | 331 | 56 | 282 | 57 | 151 | 58 | 334 | 59 | 324 |
| 60 | 324 | 61 | 208 | 62 | 175 | 63 | 64 | 64 | 327 | 65 | 208 |
| 66 | 67 | 67 | 68 | 68 | 208 | 69 | 208 | 70 | 138 | 71 | 331 |
| 72 | 324 | 73 | 175 | 74 | 53 | 75 | 254 | 76 | 338 | 77 | 276 |
| 78 | 91 | 79 | 208 | 80 | 282 | 81 | 208 | 82 | 282 | 83 | 319 |
| 84 | 85 | 85 | 208 | 86 | 310 | 87 | 324 | 88 | 208 | 89 | 90 |
| 90 | 91 | 91 | 208 | 92 | 323 | 93 | 285 | 94 | 95 | 95 | 261 |
| 96 | 276 | 97 | 98 | 98 | 324 | 99 | 282 | 100 | 288 | 101 | 102 |
| 102 | 103 | 103 | 327 | 104 | 110 | 105 | 288 | 106 | 107 | 107 | 282 |
| 108 | 276 | 109 | 110 | 110 | 324 | 111 | 110 | 112 | 288 | 113 | 114 |
| 114 | 157 | 115 | 208 | 116 | 327 | 117 | 98 | 118 | 327 | 119 | 208 |
| 120 | 208 | 121 | 110 | 122 | 324 | 123 | 208 | 124 | 125 | 125 | 208 |
| 126 | 208 | 127 | 324 | 128 | 129 | 129 | 208 | 130 | 327 | 131 | 208 |
| 132 | 208 | 133 | 28 | 134 | 135 | 135 | 136 | 136 | 324 | 137 | 138 |
| 138 | 35 | 139 | 282 | 140 | 324 | 141 | 208 | 142 | 208 | 143 | 282 |
| 144 | 324 | 145 | 146 | 146 | 282 | 147 | 148 | 148 | 208 | 149 | 208 |
| 150 | 151 | 151 | 98 | 152 | 153 | 153 | 308 | 154 | 208 | 155 | 341 |
| 156 | 157 | 157 | 208 | 158 | 324 | 159 | 208 | 160 | 208 | 161 | 98 |
| 162 | 163 | 163 | 208 | 164 | 282 | 165 | 308 | 166 | 230 | 167 | 1 |
| 168 | 285 | 169 | 208 | 170 | 171 | 171 | 208 | 172 | 208 | 173 | 208 |
| 174 | 175 | 175 | 208 | 176 | 282 | 177 | 178 | 178 | 110 | 179 | 246 |
| 180 | 208 | 181 | 282 | 182 | 324 | 183 | 282 | 184 | 208 | 185 | 324 |
| 186 | 324 | 187 | 188 | 188 | 282 | 189 | 190 | 190 | 324 | 191 | 282 |
| 192 | 193 | 193 | 135 | 194 | 35 | 195 | 28 | 196 | 282 | 197 | 307 |
| 198 | 178 | 199 | 208 | 200 | 208 | 201 | 28 | 202 | 324 | 203 | 282 |
| 204 | 208 | 205 | 206 | 206 | 282 | 207 | 208 | 208 | 91 | 209 | 324 |
| 210 | 211 | 211 | 212 | 212 | 213 | 213 | 288 | 214 | 208 | 215 | 216 |
| 216 | 282 | 217 | 246 | 218 | 335 | 219 | 276 | 220 | 282 | 221 | 222 |
| 222 | 208 | 223 | 327 | 224 | 110 | 225 | 285 | 226 | 208 | 227 | 228 |
| 228 | 208 | 229 | 324 | 230 | 327 | 231 | 232 | 232 | 208 | 233 | 282 |
| 234 | 282 | 235 | 324 | 236 | 327 | 237 | 208 | 238 | 285 | 239 | 240 |
| 240 | 331 | 241 | 285 | 242 | 324 | 243 | 208 | 244 | 309 | 245 | 107 |
| 246 | 247 | 247 | 248 | 248 | 324 | 249 | 321 | 250 | 251 | 251 | 288 |
| 252 | 135 | 253 | 254 | 254 | 327 | 255 | 208 | 256 | 208 | 257 | 341 |
| 258 | 208 | 259 | 135 | 260 | 261 | 261 | 262 | 262 | 208 | 263 | 213 |
| 264 | 208 | 265 | 327 | 266 | 208 | 267 | 268 | 268 | 269 | 269 | 208 |
| 270 | 309 | 271 | 208 | 272 | 273 | 273 | 135 | 274 | 208 | 275 | 276 |
| 276 | 277 | 277 | 324 | 278 | 279 | 279 | 208 | 280 | 281 | 281 | 282 |
| 282 | 282 | 283 | 208 | 284 | 285 | 285 | 98 | 286 | 282 | 287 | 208 |
| 288 | 310 | 289 | 324 | 290 | 282 | 291 | 309 | 292 | 208 | 293 | 294 |
| 294 | 208 | 295 | 324 | 296 | 327 | 297 | 208 | 298 | 208 | 299 | 324 |
| 300 | 208 | 301 | 285 | 302 | 324 | 303 | 282 | 304 | 282 | 305 | 282 |
| 306 | 307 | 307 | 308 | 308 | 282 | 309 | 282 | 310 | 341 | 311 | 208 |
| 312 | 313 | 313 | 331 | 314 | 282 | 315 | 282 | 316 | 282 | 317 | 282 |
| 318 | 282 | 319 | 327 | 320 | 327 | 321 | 282 | 322 | 208 | 323 | 324 |
| 324 | 325 | 325 | 324 | 326 | 327 | 327 | 282 | 328 | 329 | 329 | 282 |
| 330 | 282 | 331 | 332 | 332 | 324 | 333 | 282 | 334 | 335 | 335 | 208 |
| 336 | 337 | 337 | 338 | 338 | 208 | 339 | 340 | 340 | 341 | 341 | 342 |
| 342 | 208 | 343 | 344 | 344 | 282 | | | | | | |

**PLM** (Zhao et al., 2024) is a classifier-consistent method, which estimates the noisy class posterior with noise transition matrix to correct the label in the training. PLM crops instances to generate part-level labels, which are then modeled with a novel single-to-multiple transition matrix to capture the relationship between noisy and part-level labels.

### D.1.1 DOMAIN GENERALIZATION METHODS

**ERM** (Gulrajani & Lopez-Paz, 2021a) is the simplest baseline method, where models are simply trained on the multiple sources.

**ERM++** (Teterwak et al., 2025b) is an enhanced baseline that tunes training components to mitigate overfitting and boost generalization performance. It employs a two-stage training pipeline, explores strong initialization strategies, and investigates regularization techniques such as Model Parameter Averaging, Warm Start, Unfreezing BatchNorm, and Attention Tuning.

**MIRO** (Cha et al., 2022) is a DG framework that guides learning by maximizing mutual information between oracle representations, approximated with pre-trained models. The objective function combines empirical risk and a variational bound of the mutual information, effectively enhancing generalization ability.

**SWAD** (Cha et al., 2021) improves DG performance by finding flat minima in the loss landscape. It extends SWA with a dense sampling strategy and an overfit-aware sampling schedule, resulting in flatter minima and better generalization across domains while being robust to model selection and overfitting issues.

**Fishr** (Rame et al., 2022) is a regularization method that enforces domain invariance by matching the domain-level variances of gradients across training domains. By aligning the domain-level loss landscapes around the final weights, Fisher effectively improves out-of-distribution generalization.

**VREx** (Krueger et al., 2021) is a method that aims to reduce differences in risk across training domains to enhance robustness against extreme distributional shifts,with the assumption that domain shift is a type of variation. It proposes a penalty on the variance of training risks.

**SAGM** (Wang et al., 2023) boosts generalization across domains by finding flat regions with low loss. It minimizes the empirical risk loss, perturbed loss, and surrogate gap while performing gradient matching, efficiently guiding the model towards flat, low-loss regions, resulting in better generalization compared to SAM-like methods.

**DISAM** (Zhang et al., 2024) addresses domain-level convergence consistency in sharpness-aware minimization (SAM) by introducing a constraint to minimize variance in domain losses. This approach prevents excessive or insufficient perturbations in domains that are less or more well-optimized.

### D.2 DETAILS ON DG+LNL INTEGRATION

Algorithm 1, 2, 3, 4, 5, 6 show the detail of the integrated methods.

**Input** : Sample inputs $X = \{x_i\}_{i=1}^n$, noisy labels $\widetilde{Y} = \{\widetilde{y_i}\}_{i=1}^n$, ELR temporal ensembling
momentum $\beta$, regularization parameter $\lambda$, neural network with trainable parameters $f_\theta$
**Output:** Neural network with updated parameters $f_{\theta'}$
**for** $step \leftarrow 1$ **to** $training\_steps$ **do**
    **for** *minibatch B* **do**
        **for** *i in B* **do**
            $p_i = f_\theta(x_i)$ ; // Model prediction.
            $t_i = \beta * t_i + (1 - \beta) * p_i$ ; // Temporal ensembling.
        **end**
        loss $= -\frac{1}{|B|}\Sigma_{|B|}cross\_entropy(p_i, y_i) + \frac{\lambda}{|B|}\Sigma_{|B|}log(1- <p_i, t_i>)$ ; // ELR
        loss: cross entropy loss and regularization loss.
        Update $f_\theta$.
    **end**
    $f_{\theta'} =$ Update $f_\theta$ with ERM++ parameter averaging.
**end**

**Algorithm 1:** ERM++ + ELR Algorithm.

**Input** : Sample inputs $X = \{x_i\}_{i=1}^n$, noisy labels $\widetilde{Y} = \{\widetilde{y_i}\}_{i=1}^n$, ELR temporal ensembling
momentum $\beta$, ELR regularization parameter $\lambda 1$, MIRO regularization parameter $\lambda 2$,
MIRO mean encoder $\mu$, MIRO variance encode $\sigma$, feature extractor with trainable
parameters $f_\theta$, pretrained feature extractor with parameters $f_{\theta_0}$
**Output:** Neural network with updated parameters $f_{\theta'}$
**for** $step \leftarrow 1$ **to** $training\_steps$ **do**
    **for** *minibatch B* **do**
        **for** *i in B* **do**
            $p_i = f_\theta(x_i)$ ; // feature extractor output.
            $p_i^0 = f_{\theta_0}(x_i)$ ; // Pretrained feature extractor output.
            $t_i = \beta * t_i + (1 - \beta) * p_i$ ; // Temporal ensembling.
        **end**
        loss $= -\frac{1}{|B|}\Sigma_{|B|}cross\_entropy(p_i, y_i)$ ; // Cross entropy loss.
        loss $+= \frac{\lambda 1}{|B|}\Sigma_{|B|}log(1- <p_i, t_i>)$ ; // ELR loss with regularization
        term.
        loss $+= \frac{\lambda 2}{|B|}\Sigma_{|B|}(log(|\sigma(p_i)|) + ||p_i^0 - \mu(p_i)||_{\sigma(p_i)^{-1}}^2)$ ; // MIRO loss with
        regularization term.
        Update $f_\theta$.
    **end**
    $f_{\theta'} =$ Updated $f_\theta$.
**end**

**Algorithm 2:** MIRO + ELR Algorithm.

**Input** : Sample inputs $X = \{x_i\}_{i=1}^n$, noisy labels $\widetilde{Y} = \{\widetilde{y_i}\}_{i=1}^n$, ELR temporal ensembling momentum $\beta$, ELR regularization parameter $\lambda$, neural network with trainable parameters $f_\theta$

**Output:** Neural network with updated parameters $f_{\theta'}$

**for** $step \leftarrow 1$ **to** $training\_steps$ **do**
  **for** *minibatch $B$* **do**
    **for** *$i$ in $B$* **do**
      $p_i = f_\theta(x_i)$ ; // Model prediction.
      $t_i = \beta * t_i + (1 - \beta) * p_i$ ; // Temporal ensembling.
    **end**
    loss $= -\frac{1}{|B|}\Sigma_{|B|}cross\_entropy(p_i, y_i) + \frac{\lambda}{|B|}\Sigma_{|B|}log(1- <p_i, t_i>)$ ; // ELR
      loss: cross entropy loss and regularization loss.
    Update $f_\theta$. Decide the start $step_s$ and end $step_e$ iteration for averaging in SWAD.
  **end**
  $f_{\theta'} = \frac{1}{step_e - step_s + 1}\Sigma f_\theta$ ; // SWAD parameter averaging.
**end**

**Algorithm 3:** SWAD + ELR Algorithm.

**Input** : Sample inputs $X = \{x_i\}_{i=1}^n$, noisy labels $\widetilde{Y} = \{\widetilde{y_i}\}_{i=1}^n$, ELR temporal ensembling momentum $\beta$, ELR regularization parameter $\lambda1$, MIRO regularization parameter $\lambda2$, MIRO mean encoder $\mu$, MIRO variance encode $\sigma$, feature extractor with trainable parameters $f_\theta$, pretrained feature extractor with parameters $f_{\theta_0}$

**Output:** Neural network with updated parameters $f_{\theta'}$

**for** $step \leftarrow 1$ **to** $training\_steps$ **do**
  **for** *minibatch $B$* **do**
    **for** *$i$ in $B$* **do**
      $p_i = f_\theta(x_i)$ ; // feature extractor output.
      $p_i^0 = f_{\theta_0}(x_i)$ ; // Pretrained feature extractor output.
      $t_i = \beta * t_i + (1 - \beta) * p_i$ ; // Temporal ensembling.
    **end**
    loss $= -\frac{1}{|B|}\Sigma_{|B|}cross\_entropy(p_i, y_i)$ ; // Cross entropy loss.
    loss $+= \frac{\lambda1}{|B|}\Sigma_{|B|}log(1- <p_i, t_i>)$ ; // ELR loss with regularization
      term.
    loss $+= \frac{\lambda2}{|B|}\Sigma_{|B|}(log(|\sigma(p_i)|) + ||p_i^0 - \mu(p_i)||^2_{\sigma(p_i)^{-1}})$ ; // MIRO loss with
      regularization term.
    Update $f_\theta$. Decide the start $step_s$ and end $step_e$ iteration for averaging in SWAD.
  **end**
  $f_{\theta'} = \frac{1}{step_e - step_s + 1}\Sigma f_\theta$ ; // SWAD parameter averaging.
**end**

**Algorithm 4:** MIRO + SWAD + ELR Algorithm.

**Input** : Sample inputs $X = \{x_i\}_{i=1}^n$, noisy labels $\widetilde{Y} = \{\widetilde{y_i}\}_{i=1}^n$, MIRO regularization parameter $\lambda 2$, MIRO mean encoder $\mu$, MIRO variance encode $\sigma$, feature extractor-1 with trainable parameters $f1_\theta$, feature extractor-2 with trainable parameters $f2_\theta$, pretrained feature extractor with parameters $f_{\theta_0}$, UNICON sharpening temperature $T$, UNICON unsupervised loss coefficient $\lambda_u$, UNICON contrastive loss coefficient $\lambda_c$, , UNICON regularization loss coefficient $\lambda_r$.

**Output:** Neural network with updated parameters $f1_{\theta'}$ and $f2_{\theta'}$

**for** $step \leftarrow 1$ **to** $training\_steps$ **do**

    $D_{clean}, D_{noisy} = UNICON - Selection(X = \{x_i\}_{i=1}^n, f1_\theta, f2_\theta),$ ; // UNICON clean-noisy sample selection.

    **for** *clean minibatch* $B_{clean}$ **do**

        **for** *noisy minibatch* $B_{noisy}$ **do**

            **for** $i$ *in* $B = B_{clean} \bigcup B_{noisy}$ **do**

                $p1_i = f1_\theta(x_i)$; // feature extractor-1 output.

                $p2_i = f2_\theta(x_i)$; // feature extractor-2 output.

                $p_i^0 = f_{\theta_0}(x_i)$; // Pretrained feature extractor output.

            **end**

            $loss_1 = -\frac{1}{|B|}\Sigma_{|B|} cross\_entropy(p1_i, y_i)$; // Cross entropy loss for feature extractor-1.

            $loss_1 += \frac{\lambda 2}{|B|}\Sigma_{|B|}(log(|\sigma(p1_i)|) + ||p_i^0 - \mu(p1_i)||^2_{\sigma(p1_i)^{-1}})$; // MIRO loss with regularization term for feature extractor-1.

            $loss_2 = -\frac{1}{|B|}\Sigma_{|B|} cross\_entropy(p2_i, y_i)$; // Cross entropy loss for feature extractor-2.

            $loss_2 += \frac{\lambda 2}{|B|}\Sigma_{|B|}(log(|\sigma(p2_i)|) + ||p_i^0 - \mu(p2_i)||^2_{\sigma(p2_i)^{-1}})$; // MIRO loss with regularization term for feature extractor-2.

            $X^{weak}_{clean|B|}$ = weak-augmentation($B_{clean}$)

            $X^{weak}_{noisy|B|}$ = weak-augmentation($B_{noisy}$)

            $X^{strong}_{clean|B|}$ = strong-augmentation($B_{clean}$)

            $X^{strong}_{noisy|B|}$ = strong-augmentation($B_{noisy}$)

            Get labeled set with UNICON label refinement on clean batch.

            Get unlabeled set with UNICON pseudo label on noisy batch.

            $L_{u1}, L_{u2}$ = MixMatch on labeled and unlabeled sets ; // UNICON unsupervised loss for feature extractor-1 and extractor-2.

            Get $L_{c1}, L_{c2}$ ; // UNICON contrastive loss for feature extractor-1 and extractor-2.

            Get $L_{r1}, L_{r2}$ ; // UNICON regularization loss for feature extractor-1 and extractor-2.

            $loss_1 += \lambda_u * L_{u1} + \lambda_c * L_{c1} + \lambda_r * L_{r1}$ ; // Update UNICON loss for feature extractor-1.

            $loss_2 += \lambda_u * L_{u2} + \lambda_c * L_{c2} + \lambda_r * L_{r2}$ ; // Update UNICON loss for feature extractor-2.

            Update $f1_\theta$ and $f2_\theta$.

        **end**

    **end**

    $f1_{\theta'}$ = Updated $f1_\theta$, $f2_{\theta'}$ = Updated $f2_\theta$.

**end**

**Algorithm 5:** MIRO + UNICON Algorithm.

**Input** : Sample inputs $X = \{x_i\}_{i=1}^n$, noisy labels $\widetilde{Y} = \{\widetilde{y}_i\}_{i=1}^n$, MIRO regularization
parameter $\lambda 2$, MIRO mean encoder $\mu$, MIRO variance encode $\sigma$, feature extractor-1
with trainable parameters $f1_\theta$, feature extractor-2 with trainable parameters $f2_\theta$,
pretrained feature extractor with parameters $f_{\theta_0}$, UNICON sharpening temperature $T$,
UNICON unsupervised loss coefficient $\lambda_u$, UNICON contrastive loss coefficient $\lambda_c$, ,
UNICON regularization loss coefficient $\lambda_r$.

**Output:** Neural network with updated parameters $f1_{\theta'}$ and $f2_{\theta'}$

**for** $step \leftarrow 1$ **to** $training\_steps$ **do**

    $D_{clean}, D_{noisy} = UNICON - Selection(X = \{x_i\}_{i=1}^n, f1_\theta, f2_\theta),$; // UNICON
    `clean-noisy sample selection.`

    **for** *clean minibatch* $B_{clean}$ **do**

        **for** *noisy minibatch* $B_{noisy}$ **do**

            **for** *i in* $B = B_{clean} \bigcup B_{noisy}$ **do**

                $p1_i = f1_\theta(x_i)$; // `feature extractor-1 output.`

                $p2_i = f2_\theta(x_i)$; // `feature extractor-2 output.`

                $p_i^0 = f_{\theta_0}(x_i)$; // `Pretrained feature extractor output.`

            **end**

            $loss_1 = -\frac{1}{|B|}\Sigma_{|B|}cross\_entropy(p1_i, y_i)$; // `Cross entropy loss for`
            `feature extractor-1.`

            $loss_1 += \frac{\lambda 2}{|B|}\Sigma_{|B|}(log(|\sigma(p1_i)|) + ||p_i^0 - \mu(p1_i)||^2_{\sigma(p1_i)^{-1}})$; // `MIRO loss`
            `with regularization term for feature extractor-1.`

            $loss_2 = -\frac{1}{|B|}\Sigma_{|B|}cross\_entropy(p2_i, y_i)$; // `Cross entropy loss for`
            `feature extractor-2.`

            $loss_2 += \frac{\lambda 2}{|B|}\Sigma_{|B|}(log(|\sigma(p2_i)|) + ||p_i^0 - \mu(p2_i)||^2_{\sigma(p2_i)^{-1}})$; // `MIRO loss`
            `with regularization term for feature extractor-2.`

            $X_{clean|B|}^{weak}$ = weak-augmentation$(B_{clean})$

            $X_{noisy|B|}^{weak}$ = weak-augmentation$(B_{noisy})$

            $X_{clean|B|}^{strong}$ = strong-augmentation$(B_{clean})$

            $X_{noisy|B|}^{strong}$ = strong-augmentation$(B_{noisy})$

            Get labeled set with UNICON label refinement on clean batch.

            Get unlabeled set with UNICON pseudo label on noisy batch.

            $L_{u1}, L_{u2}$ = MixMatch on labeled and unlabeled sets ; // `UNICON`
            `unsupervised loss for feature extractor-1 and extractor-2.`

            Get $L_{c1}, L_{c2}$ ; // `UNICON contrastive loss for feature extractor-1 and extractor-2.`

            Get $L_{r1}, L_{r2}$ ; // `UNICON regularization loss for feature extractor-1 and extractor-2.`

            $loss_1 += \lambda_u * L_{u1} + \lambda_c * L_{c1} + \lambda_r * L_{r1}$ ; // `Update UNICON loss for feature extractor-1.`

            $loss_2 += \lambda_u * L_{u2} + \lambda_c * L_{c2} + \lambda_r * L_{r2}$ ; // `Update UNICON loss for feature extractor-2.`

            Update $f1_\theta$ and $f2_\theta$. Decide the start $step_s$ and end $step_e$ iteration for averaging in
            SWAD.

        **end**

    **end**

    $f1_{\theta'} = \frac{1}{step_e - step_s + 1}\Sigma f1_\theta \; f2_{\theta'} = \frac{1}{step_e - step_s + 1}\Sigma f2_\theta$ ; // `SWAD parameter averaging.`

**end**

**Algorithm 6:** MIRO + SWAD + UNICON Algorithm.

