# OpenReview forum: "Noise-Aware Generalization: Robustness to In-Domain Noise and Out-of-Domain Generalization"
_ICLR.cc/2026/Conference — ICLR 2026 Poster_

### Official Review · Reviewer_QyAy · 2025-10-28

**Soundness:** 3
**Presentation:** 3
**Contribution:** 4
**Rating:** 8
**Confidence:** 5

**Summary:**

This paper introduces Noise-Aware Generalization (NAG) to tackle noisy, diverse real-world data by improving in-domain noise handling and out-of-domain generalization. The authors show that distinguishing noise from domain shifts is challenging, and naive combinations of LNL and DG fail because domain shifts interfere with noise detection. They propose using cross-domain comparisons as a signal for identifying noise, leveraging its lack of intrinsic class features. Experiments validate that this approach significantly improves performance and offer insights for further advancing NAG.

**Strengths:**

1. The paper is well-motivated and seems to be reproducible.
2. This paper is well-organized and easy to follow. Specifically, in Sce. 3, the authors provide an in-depth analysis of the causes and implications of the proposed task, and in Sec. 4, they present concrete solutions, which are highly insightful.
3. This paper addresses real-world challenges and significantly improves task performance.

**Weaknesses:**

1. Existing works [A][B] have discussed the presence of real-world noisy labels in domain adaptation, which this paper seems to overlook. The related work section should be improved to include comparative discussion with these methods.
2. The formula proposed in Section 3.1 is explored with a toy experiment in Section 3.2. In my view, the authors could provide deeper theoretical analysis to make the argument more convincing.
3. The experiments lack some qualitative analysis to illustrate specific cases, which would help demonstrate the method’s advantages and enhance understanding of the task.
4. Currently, NAG is evaluated on synthetic noisy data. Introducing more realistic “asymmetric noise,” as studied in noisy label learning, could improve the practical applicability of NAG.
5. In my opinion, the abstract focuses almost entirely on the NAG task itself, while neglecting to highlight the insightful ideas behind the proposed method.

[A] Feng, Yanglin, et al. "ROAD: Robust unsupervised domain adaptation with noisy labels." Proceedings of the 31st ACM international conference on multimedia. 2023.

[B] Yin, Ziniu, et al. "RoDA: Robust Domain Alignment for Cross-Domain Retrieval Against Label Noise." Proceedings of the AAAI Conference on Artificial Intelligence. Vol. 39. No. 9. 2025.

**Questions:**

Please refer to the strengths and weaknesses of the paper.

---

> ### Author Response · Authors · 2025-11-20
>
> Thank you for your comments, we appreciate the time you have taken to review our paper and have revised our paper to clarify some points.  We respond to individual questions below.
>
> >1. Existing works [A][B] have discussed the presence of real-world noisy labels in domain adaptation, which this paper seems to overlook. The related work section should be improved to include comparative discussion with these methods.
>
> >[A] Feng, Yanglin, et al. "ROAD: Robust unsupervised domain adaptation with noisy labels." Proceedings of the 31st ACM international conference on multimedia. 2023.
>
> >[B] Yin, Ziniu, et al. "RoDA: Robust Domain Alignment for Cross-Domain Retrieval Against Label Noise." Proceedings of the AAAI Conference on Artificial Intelligence. Vol. 39.
>
> Thank you for the opportunity to discuss this work.  First, as you noted, these are methods [A,B] for domain adaptation, not domain generalization, i.e., their goal is to improve performance on a target domain when given a labeled source domain and unlabeled samples from a target domain.   However, in DG, the goal is to generalize to unseen distributions, thereby requiring different methods.  Additionally, they are evaluated on how well they perform on the target domain, which is similar to DG methods that evaluate performance only on unseen domains (ignoring source domain performance).  As shown in our paper, this often results in performing well only on the unseen domains, thereby limiting its applications when both ID and OOD performance is desired (e.g., MIRO in Table 7(a) obtains a 4% boost to OOD performance on CHAMMI, but at a cost to 12% ID performance).
>
> Further, when we consider adapting the ideas of these methods to our task, we find that many of them are well represented in the LNL literature.  For example, [B] separates samples into a clean and noisy set, and then aims to use the clean set to correct the noisy labels, which is also the general approach taken by relabeling methods like DISC [D] and UNICON [F].  Similarly, [A] aims to estimate label uncertainty and then re-weight samples based on this uncertainty, which is also the general approach of LNL method PLM [E].  As seen in Table 7(b), DISC, UNICON, and PLM underperform our DL4ND approach by up to 18%.
>
> The reason these methods do not transfer well is due, at least in part, to the differences in DA and DG, as there tends to only be a transfer between a single source domain and a single target domain.  Specifically, if we consider the example in Figure 1 of our paper, the main problem we raise is that in NAG we must learn to separate label noise from domain shifts.  However, in DA there is no need to solve this problem as there is only a single source domain with noise.  Thus, they are more akin to the traditional LNL task that aims to identify noise in a single domain, which is likely why similar ideas appear between them.  We simulate this type of setting by using per-domain sampling in Table 6 of our paper, effectively converting the problem into a set of single-domain noise detection tasks. However, this underperforms our approach by 2% on VLCS. This not only highlights the differences between the tasks but also illustrates the benefits of our cross-domain comparisons
>
> We’ve updated the related work section of our paper to incorporate elements of this discussion.
>
> [D] Yifan Li, Hu Han, Shiguang Shan, and Xilin Chen. Disc: Learning from noisy labels via dynamic instance-specific selection and correction. In Proceedings of the IEEE/CVF Conference on Computer Vision and Pattern Recognition, pp. 24070–24079, 2023.
>
> [E] Rui Zhao, Bin Shi, Jianfei Ruan, Tianze Pan, and Bo Dong. Estimating noisy class posterior with part-level labels for noisy label learning. In Proceedings of the IEEE/CVF Conference on Computer Vision and Pattern Recognition, pp. 22809–22819, 2024.
>
> [F] Nazmul Karim, Mamshad Nayeem Rizve, Nazanin Rahnavard, Ajmal Mian, and Mubarak Shah. Unicon: Combating label noise through uniform selection and contrastive learning. In Proceedings of the IEEE/CVF Conference on Computer Vision and Pattern Recognition, pp. 9676–9686, 2022.

---

> ### Author Response · Authors · 2025-11-20
>
> >2. The formula proposed in Section 3.1 is explored with a toy experiment in Section 3.2. In my view, the authors could provide deeper theoretical analysis to make the argument more convincing.
>
> Thank you for the suggestion.  While the toy experiment is explored in Section 3.2, many of these experiments (and more) are explored later in the paper.  In particular, in Sections 3.1 and 3.2 we use the toy dataset to show 1) that the low-loss samples can make it easier to identify noise and 2) that removing samples with uncertain labels will likely reduce performance.  Table 4 validates these observations on two real-world datasets like VLCS and CHAMMI, where w/o relabel, which removes samples rather than relabeling them, results in a 2-3% drop in most cases.  Similarly, w/o small-loss proxy also reports a 2-3% drop on both datasets.  Thus, these results validate the observations we made using RotatedMNIST in Sections 3.1 and 3.2.  Further, many alternative approaches are explored in the LNL literature, e.g., PLM and ELR (Figure 6) aim at mitigating the effect of noisy labels through regularization rather than relabeling, but perform worse than our approach.  Therefore, our experiments provide strong empirical evidence of the benefit of our approach beyond the simple toy experiments in Section 3.1 and 3.2.  We have added a reference in our paper to these results to better link the claim and supporting evidence.
>
> >3. The experiments lack some qualitative analysis to illustrate specific cases, which would help demonstrate the method’s advantages and enhance understanding of the task.
>
> We highlight that we do explore some specific cases to better understand why prior work like UNICON does not work as well in Section 5.2, where we look at questions like label accuracy on some datasets.  We also explore some other alternatives like changing the amount of samples selected for correction (Appendix A.1).  These coupled with our main results provide significant insight into why our approach works well and the challenges we face in general.  That said, we would be happy to expand this analysis with specific, actionable suggestions on what the reviewer feels is missing from our current analysis.
>
> >4. Currently, NAG is evaluated on synthetic noisy data. Introducing more realistic “asymmetric noise,” as studied in noisy label learning, could improve the practical applicability of NAG.
>
> As noted on L409 of our paper, the noise type introduced to is asymmetric noise.  We have added a note on the noise type to the caption of Tables 3 as well as modified our discussion in the experimental setup to help avoid confusion.
>
> >5. In my opinion, the abstract focuses almost entirely on the NAG task itself, while neglecting to highlight the insightful ideas behind the proposed method.
>
> Thank you for the suggestion.  In our revised paper we have adjusted the abstract to better highlight some of the ideas of our paper.  Again, we appreciate the feedback to improve our paper’s presentation.

---

> > ### Comment · Reviewer_QyAy · 2025-11-26
> >
> > Thank you to the authors for the detailed response and thorough revisions. The issues I previously raised regarding the abstract, related work, and the design and evaluation of the experiments have been addressed. I am inclined to maintain my ‘accept’ rating.

---

### Official Review · Reviewer_wQm5 · 2025-10-29

**Soundness:** 3
**Presentation:** 3
**Contribution:** 3
**Rating:** 4
**Confidence:** 4

**Summary:**

The paper studies the joint setting where data contains both domain shifts and label noise, which the authors call Noise-Aware Generalization. The idea is that existing domain generalization and noisy-label learning methods usually handle one problem at a time. The proposed method, DL4ND, uses cross-domain feature comparison to detect mislabeled samples. After a warm-up stage, it selects low-loss samples as clean, computes class-domain prototypes, and relabels high-loss samples based on the closest prototype from another domain. The method can be combined with standard DG approaches such as ERM++, SWAD, or SAGM. Experiments on seven datasets show consistent but relatively modest improvements.

**Strengths:**

1. The topic is relevant and realistic since many real-world datasets contain both domain shifts and noisy annotations.
2. The method is simple and easy to integrate into existing frameworks.
3. Experimental coverage is broad, with multiple datasets and several DG and LNL baselines.
4. The paper is well written and figures are clear.

**Weaknesses:**

1. The novelty is limited. The proposed approach mainly reuses existing ideas from noisy-label learning such as loss-based sample filtering, GMM separation, and prototype relabeling. The only new component is comparing features across domains, which is a small modification conceptually.
2. The framing of Noise-Aware Generalization as a new task feels overstated, since similar scenarios have appeared in prior DG or LNL discussions.
3. The paper lacks deeper analysis or theory explaining why cross-domain comparison works better.

**Questions:**

1. How does the proposed cross-domain comparison differ fundamentally from prototype-based noisy-label methods such as DivideMix or UNICON?
2. What motivates defining Noise-Aware Generalization as a new task rather than treating it as DG with label noise?
3. Can you provide visualization or analysis to support the claim that cross-domain comparison improves noise detection?
4. How sensitive are results to the relabeling frequency and the accuracy of domain labels?

---

> ### Author Response · Authors · 2025-11-20
>
> Thank you for your comments, we appreciate the time you have taken to review our paper and have revised our paper to clarify some points.  We respond to individual questions below.
>
> >1. The novelty is limited. The proposed approach mainly reuses existing ideas from noisy-label learning such as loss-based sample filtering, GMM separation, and prototype relabeling. The only new component is comparing features across domains, which is a small modification conceptually.
>
> Thank you for your comment, we appreciate the opportunity to discuss our paper’s novelty.  We are happy to hear that you agree that our cross-domain comparisons are novel, and we would argue that the fact this modification is easily implemented is a benefit of our approach as it means it can be quickly adapted into training workflows.  However, a claim that this limits its novelty dismisses the efforts of prior work that studied the intersection of DG and LNL, but did not offer solutions [A,B].  Specifically, if the reviewer’s argument was valid, then it would suggest that these papers studied this problem without providing any solution despite the fact that a simple approach existed (our cross-domain comparisons).  This could be possible if prior work simply noted this was a challenge that needed addressing but focused their efforts elsewhere. However, prior work published at this very conference (ICLR) has explicitly analyzed the intersection of DG and LNL (i.e., it was the main topic of the paper) [A].  I.e., this was not just some simple side analysis, but rather a thorough study directed at this problem that still did not provide any novel solutions.  If our approach was obvious *a priori*, then certainly the authors of [A,B] would have discussed it. Since they didn’t, one can only conclude that it was not an obvious solution *a priori*. Therefore, this provides evidence of **significant novelty**, as we solve a problem that prior work tried and failed to do, with a very strong contribution due, in part, to its simplicity combined with strong performance. The ease of its implementation means that it can be quickly integrated into training pipelines, increasing its potential impact.
>
> [A] Rui Qiao and Bryan Kian Hsiang Low. Understanding domain generalization: A noise robustness perspective. In The Twelfth International Conference on Learning Representations, 2024.
>
> [B] Seonguk Seo, Yumin Suh, Dongwan Kim, Geeho Kim, Jongwoo Han, and Bohyung Han. Learning to optimize domain specific normalization for domain generalization. In Computer Vision–ECCV 2020

---

> ### Author Response · Authors · 2025-11-20
>
> >2. The framing of Noise-Aware Generalization as a new task feels overstated, since similar scenarios have appeared in prior DG or LNL discussions.
> >What motivates defining Noise-Aware Generalization as a new task rather than treating it as DG with label noise?
>
> Thank you for the opportunity to discuss our task setting.  First, it is unclear whether the reviewer’s comment is well supported, as we presented NAG in several places as the intersection of DG and LNL.  For example, in the abstract of our original submission we used:
>
> “..many applications require methods that are robust to both label noise and domain shifts, which we refer to as Noise-Aware Generalization (NAG), and when…”
>
> This does not claim this is a completely new task per se, but rather explicitly names this as the intersection of these two problems.  Similar language is used elsewhere in the introduction.  We would argue that doing this simply for clarity would be sufficient justification, as it helps to reduce confusion by using different acronyms when referencing LNL, DG, and their intersection (NAG).
>
> That said, there is also ample evidence supporting the argument that they are different tasks as each (DG, LNL, and NAG) makes assumptions and has unique challenges that the others do not.  This can be seen in our summary in Fig. 2 of our paper, where prior work has one or more crucial elements missing.  For example [A,B] explore DG and LNL as noted earlier, but do not report ID performance, which limits their applications. This can be easily observed in Table 9 of the appendix where we show that DG method MIRO obtains a 4% boost to OOD accuracy over ERM on CHAMMI, the highest of any DG method, but it results in a 12% drop in ID accuracy.  Other DG methods like VREx that are explicitly touted by prior work as being robust to DG [A] also face similar fates as seen in Table 9, where they perform on par with ERM on average.
>
> Analogously, as shown in Figure 6 and Table 7(b), LNL methods, which are often originally evaluated on ID performance, perform well on ID and poorly on OOD.  Thus, we can conclude that while similar components may appear in related work, the fact that some element of our setting is missing, the methods that perform well also shift significantly.  This is due, in part, to the fact that new challenges emerge that were not present in either DG or LNL alone as highlighted in Figure 1.
>
> A hallmark of how many tasks are distinguished from each other is whether methods tend to generalize across them.  I.e., by studying either DG or LNL that we would expect to see gains in our setting, but that does not occur here as discussed above.  In fact, our DL4ND approach only emerges as a solution due to the unique properties of our task.  Thus, while LNL and DG are closely related, they are fundamentally different from NAG as evidenced by the shift in method rankings, with many older methods like ELR (published in 2020) being more effective than subsequent work.  Additionally, many methods from DG and LNL are largely ineffective in our task (performing on par or worse than ERM), which further supports the argument that direct study is needed for progress on NAG.
>
> In summary, we reiterate that our paper already framed NAG as the intersection of LNL and DG, with clarity reasons alone providing support for this naming decision.  The fact that we can't solve the problem by naive combinations of LNL+DG due to new challenges that emerge at the intersection only further supports this framing.  In addition, we have modified the discussion in our paper in an effort to further clarify this language.
>
> >3. The paper lacks deeper analysis or theory explaining why cross-domain comparison works better.
> >Can you provide visualization or analysis to support the claim that cross-domain comparison improves noise detection?
>
> Thank you for the chance to discuss our cross-domain comparisons, but it appears to ignore analysis already in Table 4 and Table 5 of our paper as well as on Table 1 of our paper.  In particular, Table 4 provides an ablation study of the components of our model, where removing our cross-domain comparisons from our model results in a 2-4% drop in performance in most settings.  Further, Table 5 reports that relabeling precision improves by up to 10% by using our cross-domain comparisons on OfficeHome and TerraIncognita, and  Table 1 reports a 23% gain in label accuracy on RotatedMNIST.  Furthermore, as we discuss in the response to your next comment, naive domain-aware alternatives to our cross-domain comparisons also underperform our method.  Thus, our analysis already provides ample empirical support for our cross-domain comparisons.  We would be happy to add to our analysis, but this would require explicit recommendations as to what is missing and a justification for why it is important.

---

> ### Author Response · Authors · 2025-11-20
>
> >1. How does the proposed cross-domain comparison differ fundamentally from prototype-based noisy-label methods such as DivideMix or UNICON?
>
> First we note that while the goal is similar, relabeling samples, the approaches of the methods differ (e.g., DividieMix and UNICON both do not use cross-domain comparisons or create proxies out of low-loss samples, unlike our approach).  That said, these differences only demonstrate a contribution if they can be experimentally validated, which we do by experimentally comparing against relabeling methods like UNICON and newer methods like DISC [C] (DivideMix is older than both UNICON and DISC).  We reproduce part of Table 7(b) below (which provides more details for summary results in Fig. 6).
>
> | VLCS | ID |  OOD  | AVG |
> | -------- | ------------------- | --------------- | --------------- |
> |  UNICON |  89.9  | 84.0    |  86.9 |
> | DISC  |  88.7   |  82.5    |  85.6  |
> | DL4ND |  **90.5**  | **86.8**           | **88.6**       |
>
> | CHAMMI | ID |  OOD  | AVG |
> | -------- | ------------------- | --------------- | --------------- |
> |  UNICON | **76.7**  | 42.0     |  59.4  |
> | DISC  |  43.3    |  41.3      |  42.3  |
> | DL4ND |  74.6  | **46.0**           | **60.3**       |
>
> As seen above, our approach outperforms UNICON and DISC on both datasets. Further, Section 5.2 provides a detailed discussion as to why methods like UNICON don’t work well.  Specifically, as seen in Figure 7, one issue is that UNICON will begin overfitting to some easy-to-learn domains, resulting in some domains being marked as largely clean and other domains being mostly relabeled.  This is analogous to the observation UNICON made about DivideMix, where they found that DivideMix would tend to relabel some categories more than others.  Their solution was to balance relabeling across categories, resulting in improved performance.  We actually extended this solution to the multi-domain setting in Table 6, where per-domain sampling ensures a uniform distribution of relabeling across domains as well.  However, it underperforms using our cross-domain comparisons by up to 2%.  Thus, naive improvements of UNICON to adapt to our setting are still insufficient to match our DL4ND approach, which validates that the differences in our method are important and make a strong contribution.
>
> [C] Yifan Li, Hu Han, Shiguang Shan, and Xilin Chen. Disc: Learning from noisy labels via dynamic instance-specific selection and correction. In Proceedings of the IEEE/CVF Conference on Computer Vision and Pattern Recognition, pp. 24070–24079, 2023.
>
> >4. How sensitive are results to the relabeling frequency?
>
> As we discuss in Section 3.1 and Section 4.2, we create our proxies using low loss samples early in training after a short warmup step without additional relabeling. If we were to update these proxies again at a later stage our analysis shows that the model would likely have begun to overfit to noisy labels (a behavior also observed in [C,D]), potentially removing some of the benefit of label correction.  This makes our approach more lightweight compared to other relabeling methods that require iterative updates (e.g., like UNICON).  As such, to address the spirit of the reviewer’s comment, we include an ablation below on the sensitivity to the warmup length (which we added to Appendix A of our paper).
>
>
> | PACS (Real-world noise) | ID | OOD  | AVG |
> | -------- | ------------------- | --------------- | --------------- |
> | w/o DL4ND | 96.3       | 85.3          | 90.8      |
> | 300 steps  | 96.7        | 84.9           | 90.8       |
> | 600 steps   | **97.3**       | **88.8**           | **93.1**       |
> | 1000 steps  |   97.2      |  88.6          | 92.9       |
>
> | TerraIncognita (20% asymmetric  noise) | ID | OOD | AVG |
> | -------- | ------------------- | --------------- | --------------- |
> | w/o DL4ND | **78.2**       | 40.4           |  59.3       |
> | 1000 steps  | 77.4       | 41.2          | 59.3       |
> | 1200 steps  | 77.1      | **43.5**           | **60.3**       |
> | 1500 steps  | 77.1      | 41.4           | 59.3      |
>
> As seen above, some datasets like PACS are not very sensitive to the exact warmup length, and a single run can be used to ablate multiple choices by storing multiple checkpoints. We made a small modification to Section 4.2 to further clarify that we only perform the relabeling once.
>
> [C] Sheng Liu, Jonathan Niles-Weed, Narges Razavian, and Carlos Fernandez-Granda. Early-learning regularization prevents memorization of noisy labels. Advances in neural information processing systems, 33:20331–20342, 2020.
>
> [D] Dongmin Choi, Sangbin Lee, EungGu Yun, Jonghyuk Baek, Frank C. Park. ELDET: Early-Learning Distillation with Noisy Labels for Object Detection. The Thirty-ninth Annual Conference on Neural Information Processing Systems, 2025

---

> ### Author Response · Authors · 2025-11-20
>
> >How sensitive are results to … the accuracy of domain labels?
>
> We would like to begin by noting that this is not an ablation that is typically performed in similar settings. For example, [A,B] did not report performance in noise from domain labels, nor it is a typical ablation performed by the *many* papers on the traditional DG task.  The reason for this is that domain labels are often considered more reliable than object labels, often due to the fact that they do not rely on human annotations.  For example, in TerraIncognita the domain labels are locations where a camera was positioned, i.e., noise cannot be easily introduced unlike for human labels.
>
> There are cases where what constitutes a domain is less heterogeneous than comparing natural images to clipart, as done in datasets like DomainNet.  For example, in CHAMMI we use the wells where an image was taken as a domain label, but these may be quite similar to each other.  This is due, in part, to the fact that differences are unintended- the biologist had intended to replicate the same process on the same type of cell for each well, but technical variations arise that result in minor differences between wells (as noted in the CHAMMI paper).  As such, some wells may be more similar to each other than they are different, and certainly are more homogenous than most datasets we evaluate on like DomainNet, PACS, and VLCS.  The only exception is TerraIncognita, which also has little variation between some domains (although arguably more than CHAMMI).  Thus, we argue the results on CHAMMI and TerraIncognita address the spirit of your remark, but in a more realistic setting, where results in Figure 6 and Table 3 show that we outperform prior work.

---

### Official Review · Reviewer_SNwE · 2025-11-01

**Soundness:** 3
**Presentation:** 3
**Contribution:** 3
**Rating:** 6
**Confidence:** 4

**Summary:**

Learning with Noisy Labels (LNL) addresses label noise and multi-source Domain Generalization (DG) handles domain shifts to boost downstream performance, but prior work mostly explores them in isolation, with limited efforts to mitigate label noise’s impact on DG.
Many applications require robustness to both label noise and distribution shifts (defined as Noise-Aware Generalization, NAG), posing challenges: LNL’s assumption that distribution shifts equal label noise fails, and DG’s neglect of label noise harms training. A naive NAG solution uses domain labels to separate shifts but wastes cross-domain information, while the proposed DL4ND improves noise detection by leveraging greater variation of noisy samples across domains. Experiments on seven diverse datasets show DL4ND significantly enhances performance.

**Strengths:**

1. This paper is easy to follow.

2. The performance of the proposed method is good. The experimentail results are relatively extensive.

**Weaknesses:**

1. It seems that there is no practical application of NAG in real world, so is it meaningful to address this new setting? The authors should discuss the potential value in real world application.

2. More related methods [1-4] in the filed of Robust Domain Adaptation under Label Noise should be reviewed and discussed.

[1] Y. Shu, Z. Cao, M. Long, and J. Wang, “Transferable curriculum for weakly-supervised domain adaptation,” in Proc. AAAI Conf. Artif. Intell., 2019, vol. 33, pp. 4951–4958.

[2] Z. Han, X. Gui, C. Cui, and Y. Yin, “Towards accurate and robust domain adaptation under noisy environments,” in Proc. 29th Int. Joint Conf. Artif. Intell., C. Bessiere, Ed., 2020, pp. 2269–2276.

[3] Y. Zuo, H. Yao, L. Zhuang, and C. Xu, “Seek common ground while reserving differences: A model-agnostic module for noisy domain adaptation,” IEEE Trans. Multimedia, vol. 24, pp. 1020–1030, 2022.

[4] Junbao Zhuo, Shuhui Wang, Qingming Huang. Uncertainty modeling for robust domain adaptation under noisy environments. IEEE Transactions on Multimedia. pp. 6157-6170. 2023.

**Questions:**

Please refer to the Weakness.

---

> ### Author Response · Authors · 2025-11-20
>
> Thank you for your comments, we appreciate the time you have taken to review our paper and have revised our paper to clarify some points.  We respond to individual questions below.
>
> >1. It seems that there is no practical application of NAG in real world, so is it meaningful to address this new setting? The authors should discuss the potential value in real world application
>
> Thank you for your comment, but we would point out that it ignores the work of hundreds, if not thousands of researchers as well as explicit examples of applications in our paper.
>
> As we state in the first paragraph of our paper and other reviewers have noted, our work is not the first to study the intersection of DG and LNL (e.g., [A,B]), but is the first to offer novel solutions.  We also provide results on three real-world datasets in diverse applications (e.g., CHAMMI representing applications in analyzing cellular images and VLCS and PACS containing natural images).  These are not datasets we created to study this problem, but simply have challenges in both DG and LNL.  Further, the other DG datasets in our paper (e.g., DomainNet, TerraIncognita) are known to have real label noise as well [C], even though we still use them in our synthetic noise experiments.
>
> Now, you could argue that VLCS, PACS, DomainNet, and OfficeHome were collected for DG and are synthetic, but this ignores the CHAMMI and TerraIncognita datasets as well as the entire field of DG that use VLCS, PACS, DomainNet, and OfficeHome to benchmark methods.  Similarly, you could argue that adding noise isn’t realistic, but that is why we provide experiments on three real-world noise datasets (Fig. 6, Tab. 2).  Additionally, adding synthetic noise is standard practice in LNL to help provide more insight into a method's ability to identify noise and control for noise ratios.  Thus, NAG’s experimental settings are similar to those in DG and LNL.  Due to this, claiming that our setup is artificial ignores the real-world examples in our paper.  Further, given the similarity in experimental setup, it also dismisses the DG literature, LNL literature, or both as artificial as well.
>
> To address this comment we’ve adjusted the discussion in the first paragraph of our paper to make these points more explicit.  However, in light of the discussion above, we reiterate that claiming our setup is artificial without any justification explaining your position lacks merit.
>
> [A] Rui Qiao and Bryan Kian Hsiang Low. Understanding domain generalization: A noise robustness perspective. In The Twelfth International Conference on Learning Representations, 2024.
>
> [B] Seonguk Seo, Yumin Suh, Dongwan Kim, Geeho Kim, Jongwoo Han, and Bohyung Han. Learning to optimize domain specific normalization for domain generalization. In Computer Vision–ECCV 2020
>
> [C] Piotr Teterwak, Kuniaki Saito, Theodoros Tsiligkaridis, Bryan A. Plummer, and Kate Saenko. Is large-scale pretraining the secret to good domain generalization? In International Conference on Learning Representations (ICLR), 2025

---

> ### Author Response · Authors · 2025-11-20
>
> >More related methods [1-4] in the filed of Robust Domain Adaptation under Label Noise should be reviewed and discussed.
>
> >[1] Y. Shu, Z. Cao, M. Long, and J. Wang, “Transferable curriculum for weakly-supervised domain adaptation,” in Proc. AAAI Conf. Artif. Intell., 2019, vol. 33, pp. 4951–4958.
>
> >[2] Z. Han, X. Gui, C. Cui, and Y. Yin, “Towards accurate and robust domain adaptation under noisy environments,” in Proc. 29th Int. Joint Conf. Artif. Intell., C. Bessiere, Ed., 2020, pp. 2269–2276.
>
> >[3] Y. Zuo, H. Yao, L. Zhuang, and C. Xu, “Seek common ground while reserving differences: A model-agnostic module for noisy domain adaptation,” IEEE Trans. Multimedia, vol. 24, pp. 1020–1030, 2022.
>
> >[4] Junbao Zhuo, Shuhui Wang, Qingming Huang. Uncertainty modeling for robust domain adaptation under noisy environments. IEEE Transactions on Multimedia. pp. 6157-6170. 2023.
>
> Thank you for the opportunity to discuss this work.  First, as you noted, these are methods [1-4] for domain adaptation, not domain generalization, i.e., their goal is to improve performance on a target domain when given a labeled source domain and unlabeled samples from a target domain.   However, in DG, the goal is to generalize to unseen distributions, thereby requiring different methods.  Additionally, they are evaluated on how well they perform on the target domain, which is similar to DG methods that evaluate performance only on unseen domains (ignoring source domain performance).  As shown in our paper, this often results in performing well only on the unseen domains, thereby limiting its applications when both ID and OOD performance is desired (e.g., MIRO in Table 7(a) obtains a 4% boost to OOD performance on CHAMMI, but at a cost to 12% ID performance).
>
> Further, when we consider adapting the ideas of these methods to our task, we find that many of them are well represented in the LNL literature.  For example, [3] searches for consistency between the predictions of a pair of networks, which is also the general approach taken by LNL method DISC [D].  Similarly, [4] aims to estimate label uncertainty and then re-weight samples based on this uncertainty, which is also the general approach of LNL method PLM [E].  As seen in Table 7(b), DISC and PLM underperform our DL4ND approach by up to 18%.
>
> The reason these methods do not transfer well is due, at least in part, to the differences in DA and DG, as there tends to only be a transfer between a single source domain and a single target domain.  Specifically, if we consider the example in Figure 1 of our paper, the main problem we raise is that in NAG we must learn to separate label noise from domain shifts.  However, in DA there is no need to solve this problem as there is only a single source domain with noise.  Thus, they are more akin to the traditional LNL task that aims to identify noise in a single domain, which is likely why similar ideas appear between them.  We simulate this type of setting by using per-domain sampling in Table 6 of our paper, effectively converting the problem into a set of single-domain noise detection tasks. However, this underperforms our approach by 2% on VLCS. This not only highlights the differences between the tasks but also illustrates the benefits of our cross-domain comparisons
>
> We’ve updated the related work section of our paper (with a longer discussion in the appendix) to incorporate elements of this discussion.
>
> [D] Yifan Li, Hu Han, Shiguang Shan, and Xilin Chen. Disc: Learning from noisy labels via dynamic instance-specific selection and correction. In Proceedings of the IEEE/CVF Conference on Computer Vision and Pattern Recognition, pp. 24070–24079, 2023.
>
> [E] Rui Zhao, Bin Shi, Jianfei Ruan, Tianze Pan, and Bo Dong. Estimating noisy class posterior with part-level labels for noisy label learning. In Proceedings of the IEEE/CVF Conference on Computer Vision and Pattern Recognition, pp. 22809–22819, 2024.

---

### Official Review · Reviewer_fTkm · 2025-11-10

**Soundness:** 2
**Presentation:** 2
**Contribution:** 2
**Rating:** 4
**Confidence:** 3

**Summary:**

This paper studies the joint problem of robustness to label noise and domain shifts, termed Noise-Aware Generalization (NAG). The authors propose DL4ND, a method that detects noisy samples by performing cross-domain comparisons among low-loss examples. The idea is that comparing samples across domains helps identify intrinsic class features and avoid confusion caused by domain-specific artifacts. Experiments on several benchmark datasets demonstrate that DL4ND improves both in-domain and out-of-domain performance over prior LNL and DG methods.

**Strengths:**

- The paper is easy to follow and logically developed.

- The proposed approach is reasonable and intuitively well-motivated, combining insights from both LNL and DG.

- Extensive experiments across multiple datasets validate the effectiveness of the method and provide detailed ablation analysis.

**Weaknesses:**

- Although the setup is novel, it appears somewhat artificial, and its practical real-world scenarios are unclear.

- The method heavily relies on the multi-domain assumption. It is uncertain how DL4ND would perform if only one source domain with noisy labels were available.

- The analysis in this manuscript (such as lines 251–258) is mostly heuristic and lacks rigorous theoretical justification or stronger empirical evidence/observation.

- The experimental evaluation could be further strengthened, for example by involving more diverse noise types.

**Questions:**

- Have the authors evaluated whether low-loss samples truly correspond to clean labels in practice?

- How robust and reliable is the clean/noisy distinction under different training dynamics or noise levels?

---

> ### Author Response · Authors · 2025-11-20
>
> Thank you for your comments, we appreciate the time you have taken to review our paper and have revised our paper to clarify some points.  We respond to individual questions below.
>
> >Although the setup is novel, it appears somewhat artificial, and its practical real-world scenarios are unclear.
>
> Thank you for your comment, but we would point out that it ignores the work of hundreds, if not thousands of researchers as well as explicit examples of applications in our paper.
>
> As we state in the first paragraph of our paper and other reviewers have noted, our work is not the first to study the intersection of DG and LNL (e.g., [A,B]), but is the first to offer novel solutions.  We also provide results on three real-world datasets in diverse applications (e.g., CHAMMI representing applications in analyzing cellular images and VLCS and PACS containing natural images).  These are not datasets we created to study this problem, but simply have challenges in both DG and LNL.  Further, the other DG datasets in our paper (e.g., DomainNet, TerraIncognita) are known to have real label noise as well [C], even though we still use them in our synthetic noise experiments.
>
> Now, you could argue that VLCS, PACS, DomainNet, and OfficeHome were collected for DG and are synthetic, but this ignores the CHAMMI and TerraIncognita datasets as well as the entire field of DG that use VLCS, PACS, DomainNet, and OfficeHome to benchmark methods.  Similarly, you could argue that adding noise isn’t realistic, but that is why we provide experiments on three real-world noise datasets (Fig. 6, Tab. 2).  Additionally, adding synthetic noise is standard practice in LNL to help provide more insight into a method's ability to identify noise and control for noise ratios.  Thus, NAG’s experimental settings are similar to those in DG and LNL.  Due to this, claiming that our setup is artificial ignores the real-world examples in our paper.  Further, given the similarity in experimental setup, it also dismisses the DG literature, LNL literature, or both as artificial as well.
>
> To address this comment we’ve adjusted the discussion in the first paragraph of our paper to make these points more explicit.  However, in light of the discussion above, we reiterate that claiming our setup is artificial without any justification explaining your position lacks merit.
>
> [A] Rui Qiao and Bryan Kian Hsiang Low. Understanding domain generalization: A noise robustness perspective. In The Twelfth International Conference on Learning Representations, 2024.
>
> [B] Seonguk Seo, Yumin Suh, Dongwan Kim, Geeho Kim, Jongwoo Han, and Bohyung Han. Learning to optimize domain specific normalization for domain generalization. In Computer Vision–ECCV 2020
>
> [C] Piotr Teterwak, Kuniaki Saito, Theodoros Tsiligkaridis, Bryan A. Plummer, and Kate Saenko. Is large-scale pretraining the secret to good domain generalization? In International Conference on Learning Representations (ICLR), 2025

---

> ### Author Response · Authors · 2025-11-20
>
> >The method heavily relies on the multi-domain assumption. It is uncertain how DL4ND would perform if only one source domain with noisy labels were available.
>
> We agree that single source DG is a worthy problem to study. However, following [C,D,E,F,G,H,I,J,K], we study a multi-source DG problem.  Note that each of these references are from papers published this year, many published after our submission, highlighting that even recent work finds this setting compelling.  While this does change potential applications, this is a very common assumption in DG, and arguably is the most ubiquitous form of the problem studied due to its recognized importance and widespread applicability. One need not go further than our own paper to see many examples of practical applications.
>
> [D] Tan Pan, Kaiyu Guo, Dongli Xu, Zhaorui Tan, Chen Jiang, Deshu Chen, Xin Guo, Brian C. Lovell, Limei Han, Yuan Cheng, Mahsa Baktashmotlagh. Minimal Semantic Sufficiency Meets Unsupervised Domain Generalization. The Thirty-ninth Annual Conference on Neural Information Processing Systems, 2025
>
> [E] Hao Zheng, Jingjun Yi, Qi Bi, Huimin Huang, Haolan Zhan, Yawen Huang, Yuexiang Li, Xian Wu, Yefeng Zheng. Learning a Cross-Modal Schrödinger Bridge for Visual Domain Generalization. The Thirty-ninth Annual Conference on Neural Information Processing Systems, 2025
>
> [F] Xavier Thomas, Deepti Ghadiyaram. What's in a Latent? Leveraging Diffusion Latent Space for Domain Generalization.  Proceedings of the IEEE/CVF International Conference on Computer Vision (ICCV), 2025
>
> [G] Yuyang Ji, Zeyi Huang, Haohan Wang, Yong Jae Lee. Customizing Domain Adapters for Domain Generalization. Proceedings of the IEEE/CVF International Conference on Computer Vision (ICCV), 2025
>
> [H] Changsong Wen, Zelin Peng, Yu Huang, Xiaokang Yang, Wei Shen. Domain Generalization in CLIP via Learning with Diverse Text Prompts. Proceedings of the IEEE/CVF Conference on Computer Vision and Pattern Recognition (CVPR), 2025
>
> [I] Cynthia Dwork, Lunjia Hu, Han Shao. How Many Domains Suffice for Domain Generalization? A Tight Characterization via the Domain Shattering Dimension. The Thirty-ninth Annual Conference on Neural Information Processing Systems, 2025
>
> [J] Dongkwan Lee, Kyomin Hwang, Nojun Kwak. Unlocking the Potential of Unlabeled Data in Semi-Supervised Domain Generalization. Proceedings of the IEEE/CVF Conference on Computer Vision and Pattern Recognition (CVPR), 2025
>
> [K] Aristotelis Ballas, Christos Diou. Gradient-Guided Annealing for Domain Generalization. Proceedings of the IEEE/CVF Conference on Computer Vision and Pattern Recognition (CVPR), 2025
>
> >The analysis in this manuscript (such as lines 251–258) is mostly heuristic and lacks rigorous theoretical justification or stronger empirical evidence/observation.
>
> Thank you for your comment, but it is not well supported, as our paper provides ample empirical evidence covering seven datasets including RotatedMNIST, VLCS, PACS, CHAMMI, DomainNet, OfficeHome, and TerraIncognita.  Additionally, Table 4 gives an ablation study where the contribution in the discussion on L251-258, which covers our cross-domain comparisons, is isolated.  There we show that removing these cross-domain comparisons reduces performance by 3-4% on VLCS and CHAMMI- two real-world noise datasets.  Table 5 also reports the labeling precision of our cross-domain comparisons on OfficeHome and TerraIncognita, where our approach provides up to a 10% improvement, and Table 1 reports a 22% gain in label accuracy on RotatedMNIST, demonstrating that our gains are due, at least in part, to more accurate label correction.  Thus, not only do we provide evidence to support our observations across the seven datasets, but provide isolated contributions of these cross-domain comparisons, resulting in very strong empirical evidence.  We’ve modified our discussion around 4.1 to refer to this supporting evidence.

---

> ### Author Response · Authors · 2025-11-20
>
> >The experimental evaluation could be further strengthened, for example by involving more diverse noise types.
>
> Thank you for your comment, but we would note that our paper already provides results with two types of noise- more than many recent papers.  Specifically, we’ll point the reviewer to Figure 2 from [L] for reference. As noted in [L], CHAMMI is an example of Dominant Noise (where the most common true label for the samples labeled for a category is not the target class).  whereas other datasets and as referenced in the experimental setup of our paper (as well as elsewhere like L409) contain the challenging and realistic asymmetric noise.  In the analysis of the noise in the real-world datasets in our paper (Appendix B), we find that VLCS and PACS are best represented as asymmetric noise as well.  Thus, our paper already represents two types of noise that includes examples of both real-world and synthetic noise.  Now, let us consider work recently published (e.g., at ICCV- after the submission deadline, and, thus, representing some of the most recent peer reviewed papers on this topic) [M,N,O,P].  Each of these papers contain either 1 (typically asymmetric) or 2 types of noise, similar or less than our paper.  While there are papers with more noise types, from these examples we can understand that the community finds work with fewer noise types valuable.
>
> What’s more, as [L] noted, Dominant Noise often violates the assumptions of many LNL methods given its high noise ratio, resulting in many methods developed for asymmetric noise performing worse than training without any kind of label correction/mitigation technique.  Yet, our approach even works on this challenging noise type.  That said, we provide results on a third type of noise, symmetric noise, below:
>
> | OfficeHome (60% symmetric  noise) | ID | OOD | AVG |
> | -------- | ------- | --------------- | --------------- |
> | ERM| 45.8       |  40.5           |  43.2       |
> | +DL4ND | **47.9**       | **49.9**           | **48.9**       |
> | SAGM | 48.6       |  40.3           |  44.4       |
> | +DL4ND | **52.0**       | **52.6**           | **52.2**       |
> | ERM++| 56.7       |  48.7           |  52.7       |
> | +DL4ND | **60.3**       | **59.4**           | **59.8**       |
>
> As can be seen above, even for symmetric noise our approach boosts performance when combined with prior work. Thus, our paper provides stronger experimental results than even work published after the submission time of our paper.  We’ve added a discussion of the varying noise types in the experimental setup at the start of Section 5.
>
> [L] Siqi Wang, Bryan A. Plummer. LNL+K: Enhancing Learning with Noisy Labels Through Noise Source Knowledge Integration. ECCV, 2024.
>
> [M] Quanjiang Li, Tingjin Luo, Jiahui Liao. Theory-Inspired Deep Multi-View Multi-Label Learning with Incomplete Views and Noisy Labels. Proceedings of the IEEE/CVF Conference on Computer Vision and Pattern Recognition (CVPR), 2025
>
> [N] Jialiang Wang, Xianming Liu, Xiong Zhou, Gangfeng Hu, Deming Zhai, Junjun Jiang, Xiangyang Ji. Joint Asymmetric Loss for Learning with Noisy Labels. Proceedings of the IEEE/CVF International Conference on Computer Vision (ICCV), 2025
>
> [O] Yuhang Li, Zhuying Li, Yuheng Jia. Boosting Class Representation via Semantically Related Instances for Robust Long-Tailed Learning with Noisy Labels. Proceedings of the IEEE/CVF International Conference on Computer Vision (ICCV), 2025
>
> [P] C Mu, Y Qu, J Yan, E Yang, C Deng. Meta-Learning Dynamic Center Distance: Hard Sample Mining for Learning with Noisy Labels. Proceedings of the IEEE/CVF International Conference on Computer Vision (ICCV), 2025

---

> ### Author Response · Authors · 2025-11-20
>
> >Have the authors evaluated whether low-loss samples truly correspond to clean labels in practice?
>
> Table 4 provides an ablation study where we measure the impact of our model components on performance.  We find that removing the low-loss proxy (and instead computing from all samples) results in a 2-3% drop in performance on VLCS and CHAMMI. Further, the cleanliness of low-loss samples has already been validated in prior work [Q,R], i.e., this is a known property that we also leverage in our paper.  While our work required modification to use the approach (i.e., using proxies rather than a classifier), as other reviewers noted, these differences are minimal.  Instead, the novelty of our paper lies in the task formulation, the combination of methods in DL4ND, and our cross-domain comparisons.  Of particular note is our proposed cross-domain comparisons to identify noisy labels, which does not appear in prior work on LNL.  We have modified our discussion in Section 3.1 to make note of these results.
>
> [Q] Sheng Liu, Jonathan Niles-Weed, Narges Razavian, and Carlos Fernandez-Granda. Early-learning regularization prevents memorization of noisy labels. Advances in neural information processing systems, 33:20331–20342, 2020.
>
> [R] Dongmin Choi, Sangbin Lee, EungGu Yun, Jonghyuk Baek, Frank C. Park. ELDET: Early-Learning Distillation with Noisy Labels for Object Detection. The Thirty-ninth Annual Conference on Neural Information Processing Systems, 2025
>
>
> >How robust and reliable is the clean/noisy distinction under different training dynamics or noise levels?
>
> Tables 3 and 5 both report performance on varying levels of synthetic noise (none, 20%, 40%). This is similar to our estimate of the noise in the real-world dataset VLCS at around 30% (see Appendix B).  However, CHAMMI represents an extreme case with much higher noise levels in some categories noted in the Dominant Noise discussion above.  In addition, we expand on our synthetic noise ratios by reporting results on 60% symmetric noise on OfficeHome above, where our approach boosts prior work by 6-8% on average.  The relative strength of performance  in this experiment also supports the observation we made in our paper that our approach's benefits tends to increase as the noise ratio increases, making any benefit of additional noise ratios negligible. In summary, our experimental settings are even more extensive than the noise ratios and types in recently published work [M,N,O,R], and our paper consistently outperforms prior work in this diverse set of applications and noise levels.

---

### Author Response · Authors · 2025-12-01
**Summary of Reviews and Discussion**

Dear AC,

Thank you for your efforts through this unusual ICLR cycle.  In an effort to assist you, we have provided a summary of the discussion across reviewers.

Overall the reviewers found that our paper is easy to read and understand (all reviewers), with a well motivated approach that addresses a real-world and important problem (Reviewers QyAy, wQm5) with a method that is easy to reproduce and integrate into other frameworks (Reviewers fTkm, wQm5, QyAy).  Further, we provide extensive experiments that validate our approach (all reviewers).


## Summary of Questions

1. Reviewers fTkm and SNwE both questioned how realistic it is to assume that the intersection of domain shifts and label noise, which we refer to as NAG, may occur concurrently.  However, this was directly contradicted by Reviewers wQm5 and QyAy who both noted that our problem setting addresses an important real-world problem.  In our response to Reviewers  fTkm and SNwE we noted that 1) our experiments contain explicit examples of NAG in real-world datasets, 2) prior work has also recognized that NAG is a setting that is practical, important, and challenging to address, and 3) that our experimental protocols are very well established from prior work in DG and LNL.

2. All reviewers noted that when discussing NAG’s challenges as well as our solution for it (i.e., the “method” section), we used results on RotatedMNIST to help illustrate the issues.  As such, they all requested verification that we show these insights transfer to more complicated real-world datasets.  For each of these questions we highlighted the relevant ablations that arose later in our paper on more challenging datasets and also adjusted our paper's method section to reference those results.  Reviewer QyAy was the only reviewer that responded to our rebuttal before discussion was disabled, and noted that it satisfied their questions and is inclined to retain their rating as "8 accept."  We highlight that Reviewer QyAy is the only one to provide a "5" on their confidence rating.

3.  Reviewers QyAy and SNwE both requested that we discuss DA+label noise methods. We provided a robust discussion on these methods in response that was added to our paper.  Notably, we highlighted the similarity of the methods proposed in the DA+label noise literature to those that appear in LNL, resulting in similar limitations. As such, representatives of DA+label noise appear in our experiments and are outperformed by our approach.

4. Reviewer wQm5 asked for experiments comparing against prior work that tries to correct label noise (e.g., DivideMix and a follow up to it, UNICON), but this appears to be due to missing key discussions and experimental results provided in our paper.  In particular, we directly compare to  UNICON and a newer method, DISC, on two real-world noise datasets (e.g., Fig. 6, Tab. 7(b)).  Further, Sec. 5.2 focuses on a lengthy discussion of why methods like UNICON do not work well in our setting, even after trying to adapt their approaches to better reflect our task (Tab. 6).  Notably, Reviewer wQm5 stated they agree that our approach of using cross-domain comparisons is novel, but they questioned its importance. However, our experiments show that our approach provides a 2-4% boost over UNICON deployed off-the-shelf, and a 2% gain on VLCS even after making changes to UNICON to better adapt to our task (see Tab. 6).  In addition, prior work, which has even been published at ICLR, directly explored a version of NAG, but proposed no solutions to it (rather just evaluating existing methods).  This supports an argument that our cross-domain comparisons were not obvious *a priori* as prior work in this topic did not propose it (or any) as a method for NAG.  Thus, this highlights that our method provides a strong contribution to the field as it solves a problem prior work tried and failed to do, and its relative simplicity is beneficial as it can be easily adapted.

5. Reviewer fTkm asked for additional experiments, but many were already present in our paper.  For example, they asked for an ablation study on whether low-loss samples in early stages of training correspond to clean labels in practice, and we provided both references from prior work highlighting this observation and an ablation in our paper that verified its benefits.  Some requested results asked us to expand upon our experiments.  We did perform additional experiments to satisfy this request, but also argued that the results are already more extensive than related work published at top-tier venues.  In other words, our experiments exceed the experimental standards of work published even after we submitted our paper, making this request unsupported by the standards used for very recent related and concurrent work.  Some experiments requested by Reviewer wQm5 also face the same criticism.

We hope the AC finds this summary useful.  We are happy to answer any further questions.

Best,

Authors of Submission 9786

---

### Meta-Review · Area_Chair_y6Ry · 2026-01-05

**Summary:**

The paper addresses Noise-Aware Generalization (NAG), a practical setting at the intersection of Domain Generalization (DG) and Learning with Noisy Labels (LNL), where models must handle both domain shifts and label noise simultaneously. The authors propose DL4ND, which utilizes cross-domain comparisons to identify and correct noisy labels, arguing that noisy samples exhibit higher variance across domains compared to clean ones.

The authors provided a comprehensive rebuttal. They clarified the distinction between NAG, DG, and DA, added comparisons to state-of-the-art LNL methods (UNICON, DISC) showing their method's superiority, and provided additional results on symmetric noise and sensitivity analysis. Reviewer QyAy explicitly confirmed their positive rating (8) following the rebuttal. While other reviewers did not respond to the rebuttal, the authors effectively addressed the technical deficiencies raised (e.g., validating the low-loss proxy assumption, justifying the cross-domain novelty). Given the extensive empirical validation across seven datasets and the clear utility of the proposed cross-domain comparison mechanism, the decision leans towards acceptance.

**Reviewer Concerns:**

The reviewers generally found the paper well-written and the problem setting well-motivated and relevant to real-world applications. However, initial concerns were raised regarding the novelty of the method (viewed by some as a recombination of LNL techniques), the artificial nature of the setup, and missing comparisons to relevant baselines in Robust Domain Adaptation (DA) and LNL (e.g., UNICON).

**Reviewer Scores:**

Reviewer's primary concerns are the lack of comparison to UNICON/DivideMix and limited novelty. The authors explicitly added the UNICON/DISC comparisons (showing improved performance) and articulated the specific failure modes of those baselines in the NAG setting, effectively countering the limited novelty argument. The authors' response detailing real-world applications (e.g., cellular imaging in CHAMMI) and the distinction from standard DA likely resolved this concern. In addition, the authors directly addressed their specific requests for additional experiments (validating low-loss samples) and provided the literature justifying the "artificial" setup.

---

### Decision · Program_Chairs · 2026-01-26

Accept (Poster)